# GMTS: GRADIENT MAGNITUDE-BASED TOKEN SELECTION IMPROVES RLVR TRAINING FOR LLM REASONING

## ABSTRACT

Reinforcement learning (RL) has recently emerged as a central paradigm for enhancing large language models' (LLMs) reasoning abilities. State-of-the-art RL with Verifiable Rewards (RLVR) methods have demonstrated remarkable effectiveness in mathematical reasoning tasks. Recent studies suggest that high-entropy tokens play an exceptionally important role in model training, since training with only the highest 20% entropy tokens yields significant performance gains. In this work, we find that while high-entropy tokens within one answer tend to correlate with large gradient magnitude, entropy alone fails to consistently reflect token importance across different answers, considering the variations in the answer-level reward signals. Based on this observation, we introduce the **G**radient **M**agnitude-based **T**oken **S**election (GMTS) method to quantify tokens. We find that training with the top 20% tokens ranked by GMTS achieves substantially better performance than entropy-based selection on well-known math benchmarks (**+1.55** on Qwen2.5-math-1.5B, **+1.33** on Qwen2.5-math-7B, **+1.85** on Qwen3-8B models). These findings indicate that GMTS provides a more refined quantification than entropy, thereby improving the performance of RLVR training. Our code is now available on https://anonymous.4open.science/r/GMTS-F8EF/

## 1 INTRODUCTION

Large language models (LLMs) have recently shown remarkable capabilities across diverse domains, such as mathematics, image quality assessment, speech, and multimodality (Achiam et al., 2023; Wu et al., 2025; Shen et al., 2025; Li et al., 2025a). Reinforcement learning (RL) has made substantial contributions to the advancement of LLMs, demonstrating outstanding effectiveness in enhancing performance (Ouyang et al., 2022; Lee et al., 2024; Shinn et al., 2023). Recently, DeepSeek introduced Group Relative Policy Optimization (GRPO) (Shao et al., 2024), a method that dispenses with the value model in favor of simple rule-based rewards and adopts group-level advantage estimation, leading to significant improvements on reasoning benchmarks. Based on GRPO, a lot of variant methods, such as Dynamic Sampling Policy Optimization (DAPO) (Yu et al., 2025) and Group Sequence Policy Optimization (GSPO) (Zheng et al., 2025), have been proposed, showing that RLVR provides a scalable and efficient paradigm for advancing the reasoning abilities of LLMs. Nevertheless, these methods adopt a uniform objective function across all tokens, without fully accounting for the fact that different tokens contribute unevenly to RLVR.

Recent studies have focused on distinguishing the token importance by designing token-level advantage functions (Yang et al., 2025; Sun et al., 2025; Deng et al., 2025a), introducing new objective functions (Cui et al., 2025; Li et al., 2025b), and filtering out low importance tokens (Wang et al., 2025c). Concurrently, several studies have also emphasized the central role of high token entropy in enhancing RLVR (Wang et al., 2025c; Cui et al., 2025; Cheng et al., 2025; Deng et al., 2025a), and controlling entropy provides a way to strengthen the RLVR reasoning ability. In these works, (Wang et al., 2025c) shows that only with 20% high entropy tokens for training yields substantially better performance across multiple mathematical reasoning tasks than training with all tokens.

*However, can entropy alone sufficiently indicate token importance?*

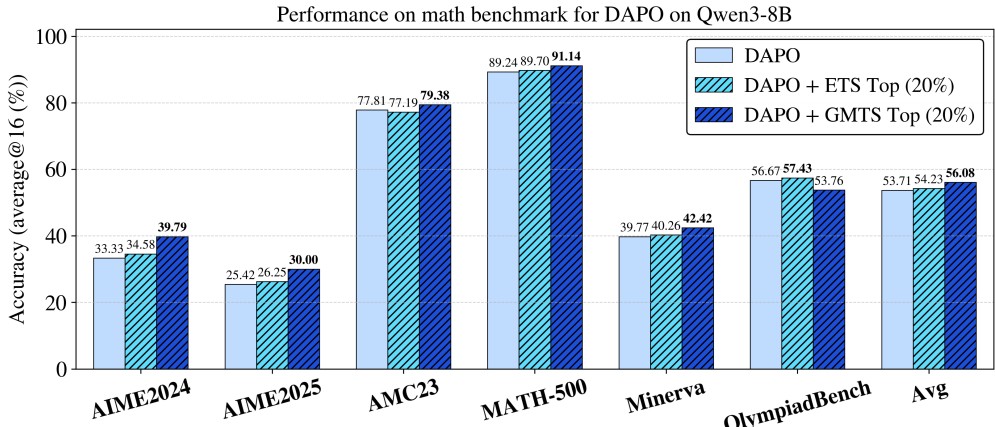

Figure 1: The average@ 16 performance of DAPO, DAPO + ETS Top (20%) and DAPO + GMTS Top (20%) on Qwen3-8B across several math benchmarks. Notably, GMTS achieves notable results compared to ETS (**+5.21** on AIME2024, **+3.75** on AIME2025, **+1.44** on MATH-500, **+2.16** on Minerva). The details are provided in Table 3.

In this work, we observe that within a single answer, high-entropy tokens tend to be associated with large gradient magnitude. However, due to variations in reward signals and sample-specific characteristics, entropy alone fails to adequately capture token importance across different answers, whereas gradient magnitude provides a more reliable quantification. Based on this, we propose the **G**radient **M**agnitude-based **T**oken **S**election (GMTS) method, and we train only on a top percentile of tokens ranked by GMTS. In contrast to **E**ntropy-based **T**oken **S**election (ETS) (Wang et al., 2025c) under the same settings, GMTS can more effectively distinguish token importance across different answers. Empirically, our experiments show that GMTS consistently improves performance by approximately **1–3** percentage points on a range of mathematical reasoning benchmarks, with gains observed across models of varying sizes. These results demonstrate the effectiveness of GMTS. Furthermore, GMTS is simple to implement and can be easily integrated into existing RLVR frameworks such as GRPO and DAPO. Since all required components are already available during standard training, GMTS introduces minimal computational overhead.

In summary, the contributions of this work are threefold:

- *Establish the Relation between token entropy and gradient magnitude.* We observe that within a single answer, tokens with high entropy correlate strongly with large gradient magnitude, which explains why entropy is an effective indicator of token importance. However, we further demonstrate that this correlation does not hold consistently across different answers when their reward signals vary significantly. This motivates the use of gradient magnitude as a more robust and reliable metric for quantifying token importance.

- *GMTS for quantifying token importance.* Building upon the relationship between entropy and gradient magnitude, we propose GMTS for quantifying token importance. GMTS effectively retains the benefits of using entropy as a signal while overcoming its key limitations in quantifying token importance across different answers. Additionally, GMTS is computationally efficient and can be easily integrated into various RLVR frameworks.

- *Empirical Studies on Mathematical Reasoning Tasks.* We test GMTS with an extensive empirical study of mathematical reasoning. The results show that GMTS, utilizing only 20% tokens, consistently outperforms the ETS baseline by 1–3 percentage points under the same selection ratio. For example, when integrated with DAPO on Qwen3-8B model, GMTS achieves improvements of **+5.21** on **AIME2024** and **+3.75** on **AIME2025**. These results not only validate the effectiveness of GMTS in enhancing RLVR performance but also highlight its potential for broader applicability across different model scales and training frameworks.

## 1.1 RELATED WORKS

Recent studies have increasingly focused on token importance, which can be roughly grouped into three complementary directions: (i) Token-level advantage estimation, which refines learning signals at the granularity of individual tokens. (ii) Objective function design, which incorporates entropy or importance-based constraints to stabilize training. (iii) Low importance token filtering, which enhances model performance and efficiency by filtering out less important tokens.

**Token-Level Advantage Estimation.** (Deng et al., 2025a) uses perplexity to dynamically adapt advantage signals, highlighting distinct roles of entropy in different training stages. (Tan & Pan, 2025) assigns entropy-weighted rewards to each token, giving higher weights to high-entropy tokens and lower weights to low-entropy ones to prevent model over-updating. At the same time, (Cheng et al., 2025) finds that high-entropy tokens are positively correlated with three types of exploration-facilitating tokens, and directly modifies the advantage with entropy. (Wang et al., 2025a) focuses on high-impact strategy tokens in reasoning and amplifies their learning signals by modifying the advantages. (Sun et al., 2025) constructs token-level contingency tables from sampled outputs to perform multinomial statistical tests, deriving importance scores that are then used to adjust token advantages.

**Objective Function Design.** (Wang et al., 2025b) proposes a clipping strategy that applies stricter constraints to low-entropy tokens while relaxing the rules for high-entropy tokens. (Cui et al., 2025) observes that decreasing entropy leads to entropy collapse and addresses this issue by dynamically adjusting entropy strength. As a follow-up, (Li et al., 2025b) identifies critical positions via high-entropy tokens, resamples them to construct branch trajectories, and jointly trains RLVR with both the branched and original trajectories. (Deng et al., 2025b) proposes Token Hidden Reward (THR) to measure each token's confidence, and uses THR to guide RLVR training. (Yao et al., 2025) conducts an in-depth exploration of diversity in RLVR reasoning and integrates RLVR's target with diversity-aware training.

**Low importance token filtering.** (Wang et al., 2025c) finds that high-entropy tokens often serve as critical forking thinking tokens, and leverages this observation by selecting only the top 20% entropy tokens for RLVR training, achieving significant performance gains. This work also lies in this direction, filtering tokens with low gradient magnitude and using remaining tokens in RLVR training.

## 2 PRELIMINARIES

### 2.1 LLM FORMULATION

A sentence is composed of several discrete units called tokens in the context of LLM. Each token corresponds to a word or subword from a finite vocabulary $\mathcal{V} = \{x^1, x^2, \ldots, x^V\}$ and $V$ denotes the vocabulary size. The process of text generation from an LLM can be formulated as a token-level Markov Decision Process (MDP), in which the LLM acts as a policy model $\pi_\theta$ parameterized by $\theta$. Given an input query $\boldsymbol{q}$, the model generates an output sequence $\boldsymbol{o}$ token by token. Specifically, at each step $t$, the model $\pi_\theta$ predicts the next token $o_t$ from a conditional probability distribution over the vocabulary $\pi_\theta(\cdot \mid \boldsymbol{q}, \boldsymbol{o}_{<t}) = [p_{t,1}, p_{t,2}, \ldots, p_{t,V}]$, where $\boldsymbol{o}_{<t}$ represents the sequence of tokens generated prior to step $t$. This iterative process continues until an end-of-sequence token is produced or the predefined maximum generation length is reached. To facilitate subsequent analysis, we define the entropy of token $o_t$ as:

$$E(o_t) := -\sum_{k=1}^{V} p_{t,k} \log p_{t,k},$$

which will be frequently used in the following sections.

### 2.2 REINFORCEMENT LEARNING WITH VERIFIABLE REWARDS

RLVR has demonstrated effectiveness in enhancing the reasoning abilities of LLMs. This section provides a brief overview of two representative RLVR methods: GRPO and DAPO.

**Group relative policy optimization (GRPO).** GRPO (Shao et al., 2024) is a variant of proximal policy optimization (PPO) (Schulman et al., 2017) that improves upon it by constructing a group advantage, thereby eliminating PPO's reliance on a value model. For any question $\boldsymbol{q}$ from the given dataset $\mathcal{D}$, GRPO samples $G$ answers $[\boldsymbol{o}_1, \boldsymbol{o}_2, \ldots, \boldsymbol{o}_G]$ from $\pi_{\text{old}}$ and collects their corresponding rewards $R = [R_1, R_2, \ldots, R_G]$ to compute the advantage $A_i = \frac{R_i - \text{mean}(R_i)}{\text{std}(R_i)}$. Within a single answer, all tokens share the same advantage $A_i$, which means that $A_{i,t} = A_i, t = 1, 2, \ldots, |\boldsymbol{o}_i|$. The optimization objective of GRPO can then be summarized as follows:

$$\max_\theta \mathbb{E}_{\boldsymbol{q} \sim \mathcal{D}, \{\boldsymbol{o}_i\} \sim \pi_{\text{old}}(\cdot|\boldsymbol{q})} \left[ \frac{1}{G} \sum_{i=1}^{G} \frac{1}{|\boldsymbol{o}_i|} \sum_{t=1}^{|\boldsymbol{o}_i|} \ell_{i,t}(\theta) \right], \tag{1}$$

Where $\ell_{i,t}(\theta)$ is the objective of token $o_{i,t}$ follows the framework of PPO:

$$\ell_{i,t}(\theta) := \min \left[ r_{i,t}(\theta) A_{i,t}, \text{clip}(r_{i,t}(\theta), 1 - \epsilon_1, 1 + \epsilon_2) A_{i,t} \right] - \beta \cdot \mathbb{D}_{KL}(\pi_\theta(o_{i,t}) | \pi_{\text{ref}}(o_{i,t})),$$

here $r_{i,t}(\theta) = \frac{\pi_\theta(o_{i,t}|\boldsymbol{q}, \boldsymbol{o}_{i,<t})}{\pi_{\text{old}}(o_{i,t}|\boldsymbol{q}, \boldsymbol{o}_{i,<t})}$, $\mathbb{D}_{KL}(\pi_\theta(o_{i,t}) | \pi_{\text{ref}}(o_{i,t}))$ is the KL divergence between the predicting probability $\pi_\theta(\cdot \mid q, \boldsymbol{o}_{<t})$ and $\pi_{\text{ref}}(\cdot \mid q, \boldsymbol{o}_{<t})$. $\pi_{\text{ref}}$ is the reference policy. $\beta$ is the parameter for the KL penalty term and $\epsilon_1, \epsilon_2$ are manually defined parameters.

**Dynamic sampling policy optimization (DAPO).** DAPO (Yu et al., 2025) is an improved version of GRPO. Comparing the difference of the training objective, the KL penalty is removed, and DAPO also introduces dynamic sampling to avoid inefficient learning that arises when all answers within a group are either entirely correct or entirely incorrect. In addition, DAPO adopts a token-level loss average and uses a higher clip parameter $\epsilon_2$ to stabilize the training process.

## 3 METHODOLOGY

Current RLVR methods, such as GRPO and DAPO, apply a uniform training objective to all tokens, neglecting their inherent differences. In this section, we analyse these differences from the perspective of gradient magnitude and reveal the relationship with token entropy.

**Gradient of each token.** As indicated in Yang et al. (2025), the gradient of each token objective $\ell_{i,t}(\theta)$ is:

$$\nabla_\theta \ell_{i,t}(\theta) = \underbrace{\left( r_{i,t}(\theta) A_{i,t} \cdot \mathbb{I}_{\epsilon_1,\epsilon_2}(r_{i,t}(\theta), A_{i,t}) + \beta \frac{\pi_{\text{ref}}(o_{i,t}|\boldsymbol{q}, \boldsymbol{o}_{i,<t})}{\pi_\theta(o_{i,t}|\boldsymbol{q}, \boldsymbol{o}_{i,<t})} - \beta \right)}_{=: \omega_{i,t}(\theta)} \nabla_\theta \log \pi_\theta(o_{i,t}|\boldsymbol{q}, \boldsymbol{o}_{i,<t})$$

where $\mathbb{I}_{\epsilon_1,\epsilon_2}(r_{i,t}(\theta), A_{i,t})$ is the indictor function with parameter $\epsilon_1, \epsilon_2$ defined as follows:

$$\mathbb{I}_{\epsilon_1,\epsilon_2}(r_{i,t}(\theta), A_{i,t}) = \begin{cases} 0 & \text{if } A_{i,t} > 0, \ r_{i,t}(\theta) > 1 + \epsilon_2, \\ 0 & \text{if } A_{i,t} < 0, \ r_{i,t}(\theta) < 1 - \epsilon_1, \\ 1 & \text{otherwise.} \end{cases}$$

The coefficient of gradient $\omega_{i,t}(\theta)$ is approximately $A_{i,t}$. For example, in the first mini batch traning of each step in DAPO framework, the KL term is omitted and $\pi_\theta = \pi_{\text{old}}$, then $\omega_{i,t}(\theta)$ is exactly $A_{i,t}$.

**Relation between token gradient magnitude and entropy.** As illustrated in the left panel of Fig. 3, the magnitude of token-level log-prob gradient $\nabla_\theta \log \pi_\theta(o_{i,t}|\boldsymbol{q}, \boldsymbol{o}_{<t})$ exhibits an approximately linear relationship with token entropy when both axes are plotted on a logarithmic scale. To interpret this phenomenon, we analyze the gradient formulation. Consider a query input $\boldsymbol{q}$ and an output sequence $\boldsymbol{o}$, we focus on the $t$-th token $o_t$. Let $\boldsymbol{z}_t = [z_{t,1}, z_{t,2}, \ldots, z_{t,V}]$ denote the logits output from the final layer of the LLM at time $t$, and let $\pi_\theta(\cdot | \boldsymbol{q}, \boldsymbol{o}_{<t}) = \text{Softmax}(\boldsymbol{z}_t) := [p_{t,1}, p_{t,2}, \ldots, p_{t,V}]$ be the predicted probability distribution over the vocabulary for token $o_t$. The expectation of $\ell_1$-norm of the gradient $\nabla_{\boldsymbol{z}_t} \log \pi_\theta(o_t|\boldsymbol{q}, \boldsymbol{o}_{<t})$ is given by:

$$\mathbb{E}_{o_t \sim \pi_\theta(\cdot|\boldsymbol{q}, \boldsymbol{o}_{<t})} \left[ \| \nabla_{\boldsymbol{z}_t} \log \pi_\theta(o_t|\boldsymbol{q}, \boldsymbol{o}_{<t}) \|_1 \right] = 2 \sum_{k=1}^{V} p_{t,k}(1 - p_{t,k}). \tag{2}$$

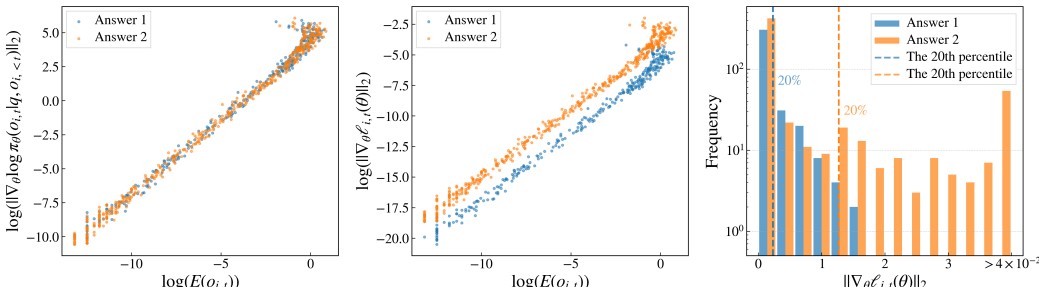

Figure 2: We use DAPO to train Qwen2.5-math-1.5B on MATH-12K dataset under the setting of $G = 16$. At training step 100, we selected two answers with different advantages in one question group, and computed the gradients $\nabla_\theta \ell_{i,t}(\theta)$, $\nabla_\theta \log \pi_\theta(o_{i,t}|\boldsymbol{q}, \boldsymbol{o}_{<t})$, and entropy $E(o_{i,t})$ of all the tokens. With these collected data, we plotted entropy against the token-level log-probability gradients under the log scale **(left)**, entropy against the true gradient magnitude under the log scale **(middle)**, and the distributions of true gradient magnitude together with their 20th percentile boundaries **(right)**.

A detailed derivation of (2) is provided in Appendix. A.2. Meanwhile, the entropy of token $o_t$ can be approximated as:

$$E(o_t) = -\sum_{k=1}^{V} p_{t,k} \log p_{t,k} \approx \sum_{k=1}^{V} p_{t,k}(1 - p_{t,k}), \tag{3}$$

where (3) is approximated through Gini index (Martino & Elvira, 2025; Tatti, 2025). Let $\frac{\partial \boldsymbol{z}_t}{\partial \theta}$ denote the Jacobian matrix of the logits $\boldsymbol{z}_t$ with respect to parameters $\theta$. Since $\nabla_\theta \pi_\theta(o_t|\boldsymbol{q}, \boldsymbol{o}_{<t}) = \frac{\partial \boldsymbol{z}_t}{\partial \theta} \cdot \nabla_{\boldsymbol{z}_t} \pi_\theta(o_t|\boldsymbol{q}, \boldsymbol{o}_{<t})$, (2) and (3) togather indicate that tokens with higher entropy generally correspond to larger gradients of the log-probability. These results provide more insights into why high-entropy tokens tend to contribute more significantly in the RLVR training process Wang et al. (2025c).

**Gradient Magnitude-based Token Selection method.** However, the coefficients $\omega_{i,t}(\theta)$ in the token gradients can exhibit substantial variability and are particularly influenced by the advantage values derived from answer-level reward signals. As shown in the middle and right panels of Fig. 2, tokens belonging to answers with different advantages display notable differences in gradient magnitudes, along with distinct percentile distributions. These observations suggest that entropy alone is insufficient to reflect a token's contribution in RLVR training. To address this limitation, we propose the GMTS method, which assesses token importance directly from the gradient magnitude. Since directly computing the true gradient norm is computationally intractable during RL training, we approximate $\nabla_\theta \log \pi_\theta(o_{i,t}|\boldsymbol{q}, \boldsymbol{o}_{i,<t})$ using token entropy. And we define the GMTS ratio as:

$$\delta_{i,t}(\theta) = |E(o_{i,t}) \cdot \omega_{i,t}(\theta)|$$

which serves as a tractable measure of token importance. Based on that, the training objective of GMTS is as follows:

$$\max_\theta \mathbb{E}_{\boldsymbol{q}\sim\mathcal{D}, \{\boldsymbol{o}_i\}\sim\pi_{\text{old}}(\cdot|\boldsymbol{q})} \left[ \frac{1}{G}\sum_{i=1}^{G} \frac{1}{|\boldsymbol{o}_i|} \sum_{t=1}^{|\boldsymbol{o}_i|} \mathbb{I}[\delta_{i,t}(\theta) \geq \tau_\rho] \cdot \ell_{i,t}(\theta) \right], \tag{4}$$

where $\mathbb{I}(\cdot)$ is the indicator function that evaluates to 1 if the condition inside holds and 0 otherwise. $\rho$ is a predefined ratio specifying the top proportion to be selected, and $\tau_\rho$ is the corresponding threshold such that only tokens with $\delta_{i,t}(\theta) \geq \tau_\rho$, comprising the top-$\rho$ fraction of all tokens in the batch, are used to compute the gradient.

Compared to the ETS method Wang et al. (2025c), the primary distinction of our method lies in the token importance measurement. Both these two approaches utilize only the top-$\rho$ fraction of tokens for training. From a computational perspective, calculating $\omega_{i,t}(\theta)$ involves the advantage signal $A_{i,t}$ and the predicted probability from $\pi_\theta, \pi_{\text{old}}$ and $\pi_{\text{ref}}$ for each token, all those quantities are already abailable from text generating and are requisite in the conventional RLVR training. Therefore, the additional computational overhead introduced by GMTS is minimal. Moreover, GMTS method

can be readily integrated into other RLVR training frameworks, such as DAPO, with the only difference being the specific formulation of the coefficient $\omega_{i,t}(\theta)$. By leveraging only a small subset of influential tokens for RLVR updates, GMTS offers the potential for more efficient RLVR training.

## 4 NUMERICAL EXPERIMENTS

### 4.1 TRAINING AND EVALUATION SETTING

**Training Settings.** We evaluate our approach within the verl framework [1] under both the DAPO and GRPO algorithms. We further develop an independent implementation within [2] that supports efficient training with a limited number of GPUs. Our main experiments are conducted on three models: **Qwen2.5-math-1.5B**, **Qwen2.5-math-7B**, and **Qwen3-8B**. This selection allows us to examine model behavior across different parameter scales, thereby validating the generality and scalability of GMTS. Some key training parameters are summarized below, while other details are provided in Appendix A.3.

- For **Qwen2.5-math-1.5B** and **Qwen2.5-math-7B** we use the complete MATH-12K dataset (Lightman et al., 2023) as the training set. The group size $G = 16$. The maximum response length is set to 2048 tokens, and the maximum prompt length is 1024 tokens, with both training and mini-batch sizes set to 64. Here we set the global learning rate to $3 \times 10^{-5}$.
- For **Qwen3-8B**, we follow (Wang et al., 2025c) with most configurations unchanged, except that the maximum response length and prompt length are set to 4096 and 1024 tokens due to computational limits.

We adapt the Qwen-math template as the chat format for all experiments:

```
<|im_start|>system\nPlease reason step by step, and put your final answer
 within \boxed{}.<|im_end|>\n<|im_start|>user\n"<|im_end|>\n<|im_start|>
assistant\n
```

For reward design, we extract the answer in the \boxed{} and compare it with the ground-truth result, assigning a reward of 1 for correctness and 0 otherwise.

**Evaluation Settings.** Our experiments are conducted on six math benchmark datasets: **AIME2025**, **AIME2024**, **AMC23**, **MATH-500**, **Minerva**, and **OlympiadBench**. For Qwen2.5-math-1.5B and Qwen2.5-math-7B, since **AIME2025** is particularly challenging, small models (1.5B and 7B) achieve very low accuracy (below 10%), we therefore exclude these results from this benchmark. For each question in each dataset, we generate 16 candidate answers and report the average accuracy as *average@16*. All generations are performed with the temperature of $T = 1.0$, consistent with the training settings.

### 4.2 RESULTS ON QWEN2.5-MATH-1.5B AND 7B

We implement ETS/GMTS on GRPO/DAPO, and apply them to the Qwen2.5-math-1.5B and 7B model. We give an example for DAPO on Qwen2.5-math-7B with ETS and GMTS in Fig. 3, with the curves under a sliding window of 20. We also present all the detailed results in Tables 1–2.

From Tables 1–2, we observe that GMTS Top 20% outperforms ETS at the same level under both DAPO and GRPO (1.5B-DAPO: **+1.55**, 1.5B-GRPO: **+1.30**, 7B-DAPO: **+1.33**, 7B-GRPO: **+3.41**). Moreover, GMTS remains highly competitive even at the Top 10% level (1.5B-DAPO: **+0.88**, 1.5B-GRPO: **+0.88**, 7B-DAPO: **+0.58**, 7B-GRPO: **+2.84**). This not only demonstrates that GMTS is more effective, but also indicates that it can achieve superior performance with fewer selected ratio of tokens.

From Fig. 3(a), we observe that GMTS attains a clear advantage in average accuracy on AIME2024, reflecting its ability to sustain higher reward signals during training. From Fig. 3(b), the entropy curve of GMTS lies between those of ETS and DAPO, indicating that GMTS strikes a balance: being more explorative than DAPO while maintaining greater stability than ETS.

---

[1] https://github.com/volcengine/verl

[2] https://github.com/policy-gradient/GRPO-Zero

Table 1: Evaluation on five math-reasoning benchmarks (ignore AIME2025) for ETS/GMTS with DAPO/GRPO on **Qwen2.5-math-1.5B** under *average@16* accuracy (%).

|  | AIME2024 | AMC23 | MATH-500 | Minerva | OlympiadBench | Avg. |
|---|---|---|---|---|---|---|
| **Qwen2.5-math-1.5B** | 2.70 | 18.75 | 23.40 | 6.20 | 14.10 | 13.03 |
| DAPO | 14.17 | 50.62 | **73.85** | 29.47 | 37.17 | 41.06 |
| + ETS Top (20%) | 14.17 | 48.44 | 72.25 | 29.01 | 36.16 | 40.01 |
| **+ GMTS Top (20%)** | **14.58** | 50.63 | 73.83 | **30.51** | **38.23** | **41.56** |
| **+ GMTS Top (10%)** | 13.33 | **52.19** | 73.53 | 29.72 | 35.68 | 40.89 |
| GRPO | 10.63 | 48.12 | 71.59 | 28.81 | 34.90 | 38.81 |
| + ETS Top (20%) | 12.71 | 48.89 | 72.39 | 28.41 | 35.54 | 39.59 |
| **+ GMTS Top (20%)** | 11.88 | **53.33** | **73.06** | **30.11** | **36.09** | **40.89** |
| **+ GMTS Top (10%)** | **15.00** | 49.84 | 72.24 | 29.60 | 35.65 | 40.47 |

Table 2: Evaluation on five math-reasoning benchmarks (ignore AIME2025) for ETS/GMTS with DAPO/GRPO on **Qwen2.5-math-7B** under *average@16* accuracy (%).

|  | AIME2024 | AMC23 | MATH-500 | Minerva | OlympiadBench | Avg. |
|---|---|---|---|---|---|---|
| **Qwen2.5-math-7B** | 7.29 | 21.09 | 34.45 | 6.09 | 10.92 | 15.97 |
| DAPO | 18.12 | 62.50 | 80.91 | 36.49 | 44.35 | 48.47 |
| + ETS Top (20%) | 20.00 | 63.75 | 80.04 | 36.72 | 43.56 | 48.81 |
| **+ GMTS Top (20%)** | **25.00** | **65.16** | 80.40 | **37.39** | 42.75 | **50.14** |
| **+ GMTS Top (10%)** | 20.00 | 63.44 | **81.61** | 37.34 | **44.56** | 49.39 |
| GRPO | **27.08** | 62.34 | 79.76 | 35.18 | 42.87 | 49.45 |
| + ETS Top (20%) | 19.17 | 60.67 | 77.72 | 33.65 | 40.96 | 46.43 |
| **+ GMTS Top (20%)** | 23.33 | **64.83** | **81.13** | **36.32** | **43.60** | **49.84** |
| **+ GMTS Top (10%)** | 23.54 | 63.91 | 80.32 | 36.19 | 42.41 | 49.27 |

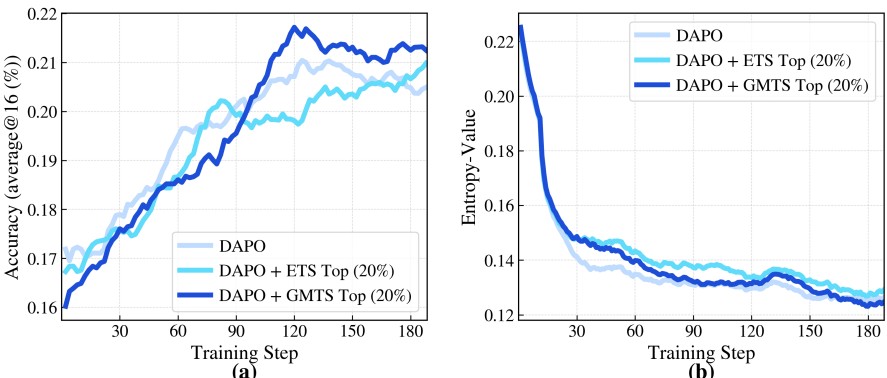

Figure 3: Training curves of DAPO, DAPO + ETS Top (20%), and DAPO + GMTS Top (20%) on Qwen2.5-math-7B under $G = 16$. (a) shows the AIME2024 accuracy during training with a sliding window of 20, (b) shows the average token entropy under the same sliding window.

## 4.3 RESULTS ON QWEN3-8B

We conduct additional experiments on the more powerful Qwen3-8B model, following the settings and the experimental results of (Wang et al., 2025c) as the baseline for comparison. The performance results are in Table 3.

It can be observed that GMTS consistently delivers notable improvements over ETS across a wide range of math benchmarks, achieving (**+5.21** on AIME2024, **+3.75** on AIME2025, **+2.19** on AMC23, **+1.44** on MATH-500, and **+2.16** on Minerva). Taken together, these results yield an overall average improvement of **1.85**, underscoring the robustness and generality of GMTS across

Table 3: Evaluation on six math-reasoning benchmarks for ETS/GMTS with DAPO on **Qwen3-8B** under *average@16* accuracy (%).

|  | AIME2024 | AIME2025 | AMC23 | MATH-500 | Minerva | OlympiadBench | Avg. |
|---|---|---|---|---|---|---|---|
| **Qwen3-8B** | 3.75 | 6.04 | 33.91 | 62.99 | 22.29 | 23.66 | 25.44 |
| DAPO | 33.33 | 25.42 | 77.81 | 89.24 | 39.77 | 56.67 | 53.71 |
| + ETS Top (20%) | 34.58 | 26.25 | 77.19 | 89.70 | 40.26 | **57.43** | 54.23 |
| **+ GMTS Top (20%)** | **39.79** | **30.00** | **79.38** | **91.14** | **42.42** | 53.76 | **56.08** |

tasks of varying difficulty. Such consistent advantages highlight not only the effectiveness of GMTS in capturing task-specific nuances but also its strong potential to scale favorably to larger and more complex models.

## 4.4 ABLATION STUDY

**Tokens with lower gradient magnitude contribute less.** Following (Wang et al., 2025c), we also conduct bottom selection. We set the clipping ratios to 80% and 90% and perform the experiments with DAPO on Qwen2.5-math-1.5B. The results are shown in Fig. 4.

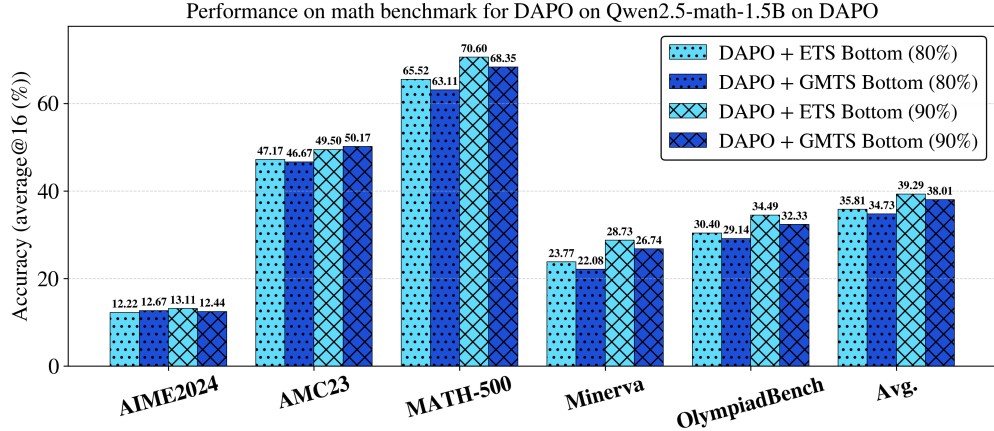

Figure 4: The average@ 16 performance of DAPO, DAPO + ETS Bottom (80%, 90%) and DAPO + GMTS Top (80%, 90%) on Qwen2.5-math-1.5B.

With the completed results provide in Appendix A.1, Table 5, and further illustrat in Fig. 4, we observe that GMTS + Bottom consistently underperforms compared to ETS + Bottom under the same selected ratios (showing an average decrease of **-1.08** at the 80% ratio and **-1.28** at the 90% ratio). These findings indicate that tokens with lower gradient magnitude contribute only marginally to effective RLVR training. In contrast, low-entropy tokens still retain useful signals, thereby highlighting that entropy-based token selection may not represent the most effective strategy.

**Effect of varying selected ratios.** To compare GMTS and ETS under different selection ratios, we evaluate both DAPO and GRPO on Qwen2.5-math-1.5B with ratios 0.1, 0.2, 0.5, 0.7, and 0.9, with the results reported in Table 4 and Fig. 5 (left and middle). We observe that GMTS achieves higher average performance than ETS in most settings (**+2.19** on DAPO-90%, **+0.80** on DAPO-70%, **+0.40** on DAPO-50%, **+1.55** on DAPO-20%, **+2.96** on DAPO-10%) (**+1.09** on GRPO-90%, **+0.88** on GRPO-70%, **+1.30** on GRPO-20%, **+3.10** on GRPO-10%). These results demonstrate that the advantages of GMTS are not confined to a single regime but extend across a broad spectrum of selected ratios, suggesting that GMTS provides a more stable and reliable improvement than ETS under varying training conditions.

**Model scaling amplifies GMTS gains on AIME2024.** The improvement of GMTS on AIME 2024 becomes more pronounced with increasing model size, from the right panel of Figure 5, as the model scales from 1.5B (Qwen2.5-math-1.5B) to 8B (Qwen3-8B), the avg@16 accuracy gain rises

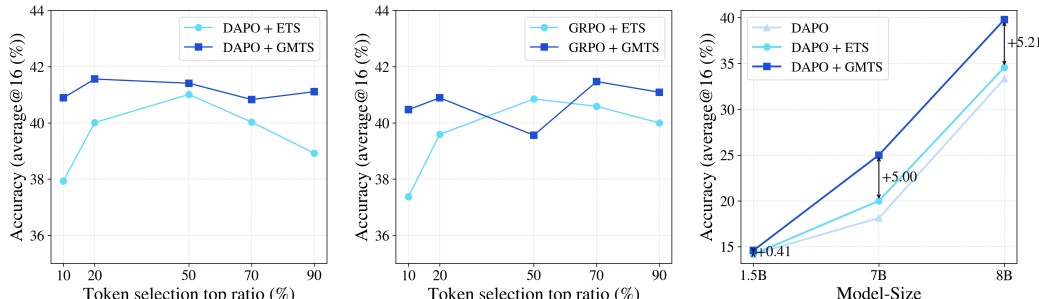

Figure 5: The average@16 overall performance of varying selected ratios for DAPO, DAPO + ETS Top and DAPO + GMTS Top on Qwen2.5-math-1.5B (left and middle). Avg@16 accuracy on AIME2024 across different model sizes. Results for TES and GMTS are reported with the top-20% selection.

from **+0.41** to **+5.20**, this phenomenon is consistent with (Wang et al., 2025c), further indicating the greater potential of GMTS on larger models.

Table 4: Evaluation on five math-reasoning benchmarks for ETS/GMTS with DAPO/GRPO on **Qwen2.5-math-1.5B** under *average@16* accuracy (%) in different selected ratios.

|  | AIME2024 | AMC23 | MATH-500 | Minerva | OlympiadBench | Avg. |
|---|---|---|---|---|---|---|
| **Qwen2.5-math-1.5B** | 2.70 | 18.75 | 23.40 | 6.20 | 14.10 | 13.03 |
| DAPO | 14.17 | 50.62 | 73.85 | 29.47 | 37.17 | 41.06 |
| + ETS Top (90%) | **13.96** | 51.72 | 73.97 | **30.36** | 24.58 | 38.92 |
| **+ GMTS Top (90%)** | 11.04 | **52.34** | **74.20** | **30.36** | **37.61** | **41.11** |
| + ETS Top (70%) | **12.08** | 48.44 | **74.07** | 29.26 | 36.31 | 40.03 |
| **+ GMTS Top (70%)** | 11.46 | **52.50** | 74.00 | 28.80 | **37.40** | **40.83** |
| + ETS Top (50%) | 12.50 | 52.81 | 73.22 | 29.55 | **36.97** | 41.01 |
| **+ GMTS Top (50%)** | **13.54** | **53.44** | **73.66** | **29.89** | 36.50 | **41.41** |
| + ETS Top (10%) | 12.71 | 46.88 | 70.03 | 27.08 | 32.93 | 37.93 |
| **+ GMTS Top (10%)** | **13.33** | **52.19** | **73.53** | **29.72** | **35.68** | **40.89** |
| GRPO | 10.63 | 48.12 | 71.59 | 28.81 | 34.90 | 38.81 |
| + ETS Top (90%) | 11.56 | 50.00 | **72.73** | 29.61 | **36.07** | 40.00 |
| **+ GMTS Top (90%)** | **15.33** | **51.00** | 72.68 | **30.51** | 35.91 | **41.09** |
| + ETS Top (70%) | 11.88 | 50.62 | 73.51 | 30.31 | 36.61 | 40.59 |
| **+ GMTS Top (70%)** | **12.92** | **52.50** | **74.22** | **30.61** | **37.11** | **41.47** |
| + ETS Top (50%) | **12.92** | **50.78** | **73.46** | **29.25** | **37.86** | **40.85** |
| **+ GMTS Top (50%)** | 10.63 | 49.38 | 72.95 | 28.75 | 36.09 | 39.56 |
| + ETS Top (10%) | 10.63 | 45.94 | 70.00 | 27.30 | 32.97 | 37.37 |
| **+ GMTS Top (10%)** | **15.00** | **49.84** | **72.24** | **29.60** | **35.65** | **40.47** |

## 5 CONCLUSION

In this work, we propose Gradient Magnitude-based Token Selection (GMTS) as a further generalization of Entropy-based Token Selection (ETS). While gradient magnitude correlates with entropy within an answer, entropy alone fails to capture token importance across answers. GMTS addresses this limitation and achieves consistent improvements: it significantly outperforms ETS at the top-20% ratio, remains superior even at 10%, and shows robustness across multiple selection ratios and model scales (Qwen2.5-math-1.5B, 7B, and Qwen3-8B). These results establish GMTS as a principled and efficient framework for RLVR.

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

# A    APPENDIX

## A.1    MORE EXPERIMENTAL RESULTS

Table 5: Evaluation on five math-reasoning benchmarks (ignore AIME2025) for ETS/GMTS with DAPO/GRPO on **Qwen2.5-math-1.5B** under *average@16* accuracy (%).

|                    | AIME2024 | AMC23 | MATH-500 | Minerva | OlympiadBench | Avg. |
|--------------------|----------|-------|----------|---------|---------------|------|
| **Qwen2.5-math-1.5B** | 2.70  | 18.75 | 23.40    | 6.20    | 14.10         | 13.03 |
| **DAPO**           | 18.12    | 62.50 | 80.91    | 36.49   | 44.35         | 48.47 |
| + ETS Bottom (80%) | 12.22    | 47.17 | 65.52    | 23.77   | 30.40         | 35.81 |
| + ETS Bottom (90%) | 13.11    | 49.50 | 70.60    | 28.73   | 34.49         | 39.29 |
| + GMTS Bottom (80%) | 12.67   | 46.67 | 63.11    | 22.08   | 29.14         | 34.73 |
| + GMTS Bottom (90%) | 12.44   | 50.17 | 68.35    | 26.74   | 32.33         | 38.01 |

## A.2    DERIVATION OF TOKEN GRADIENT

Consider a query input $q$ and an output sequence $o$, we focus on the gradient magnitude and entropy of $t$-th token $o_t$. Let $z_t = [z_{t,1}, z_{t,2}, \ldots, z_{t,V}]$ denote the logits output of the LLM at time $t$. And $\pi_\theta(\cdot \,|\, q, o_{<t}) = \mathrm{Softmax}(\frac{z_t}{T}) := [p_{t,1}, p_{t,2}, \ldots, p_{t,V}]$ be the predicted probability distribution over the vocabulary for token $o_t$. Here we set the temperature $T = 1$:

$$p_{t,k} = \frac{\exp(z_{t,k})}{\sum_{j=1}^{V} \exp(z_{t,j})}.$$

Thus, for the partial derivative $\frac{\partial p_{t,k}}{\partial z_{t,l}}$, if $k = l$ then

$$\frac{\partial p_{t,k}}{\partial z_{t,k}} = \frac{\exp(z_{t,k})}{\sum_{j=1}^{V} \exp(z_{t,j})} - \left( \frac{\exp(z_{t,k})}{\sum_{j=1}^{V} \exp(z_{t,j})} \right)^2 = p_{t,k} - p_{t,k}^2,$$

if $k \neq l$ then

$$\frac{\partial p_{t,k}}{\partial z_{t,l}} = -\frac{\exp(z_{t,k})}{\sum_{j=1}^{V} \exp(z_{t,j})} \cdot \frac{\exp(z_{t,l})}{\sum_{j=1}^{V} \exp(z_{t,j})} = -p_{t,k} \cdot p_{t,l},$$

Thus, the partial derivative can be summarized as

$$\frac{\partial p_{t,k}}{\partial z_{t,l}} = p_{t,k}(\mathbb{I}(k = l) - p_{t,l})$$

Where $\mathbb{I}(k = l)$ takes the value 1 when $k = l$ and 0 otherwise. Then the $\ell_1$-norm of $\nabla_{z_t} p_{t,k}$ is

$$\|\nabla_{z_t} p_{t,k}\|_1 = \sum_{l=1}^{V} \left| \frac{\partial p_{t,k}}{\partial z_{t,l}} \right| = 2 \cdot p_{t,k}(1 - p_{t,k}),$$

thus the expectation

$$\mathbb{E}_{o_t \sim \pi_\theta(\cdot \,|\, q, o_{<t})} \left[ \|\nabla_{z_t} \log \pi_\theta(o_t | q, o_{<t})\|_1 \right] = \mathbb{E}_{p_{t,k} \sim p_t} \left[ \left\| p_{t,k}^{-1} \cdot \nabla_{z_t} p_{t,k} \right\|_1 \right]$$

$$= 2 \cdot \sum_{k=1}^{V} p_{t,k}(1 - p_{t,k})$$

## A.3    TRAINING DETAILS

For fairness and reproducibility, we fix the training seed to **0** in all the experiments, which are performed on eight NVIDIA H100 GPUs, each equipped with 80 GB of memory.

**Hyperparameters settings for Qwen2.5-math-1.5B and Qwen2.5-math-7B.**

For the Qwen2.5-math-1.5B and Qwen2.5-math-7B models, the training configurations are as follows: the maximum response length is set to 2048 tokens, and the maximum prompt length to 1024 tokens. Each training batch contained 64 questions, with a mini-batch size of 64. For each question, 16 rollouts are sampled with a temperature of 1.0. The clipping parameters $\epsilon_1$ and $\epsilon_2$ are set to 0.2 and 0.28. The learning rate is fixed at $3 \times 10^{-5}$ without applying any learning rate warm-up or scheduling

**Hyperparameters settings for Qwen3-8B.**

Due to computational constraints, we are unable to fully adapt the settings of (Wang et al., 2025c). and thus adjust the maximum output length. Specifically, we set the maximum response length to 4096 tokens and the maximum prompt length to 1024 tokens. Each training batch contained 512 questions and a mini-batch size of 32, resulting in 16 gradient steps per training batch. For each question, 16 rollouts are sampled with a temperature of 1.0. The clipping parameters $\epsilon_1$ and $\epsilon_2$ are set to 0.2 and 0.28. The learning rate is fixed at $1 \times 10^{-6}$ without applying any learning rate warm-up or scheduling

### A.4 AN EXAMPLE FOR ETS TOP (20%) AND GMTS (20%) WITH DAPO ON QWEN3-8B UNDER AIME2025

In DAPO on Qwen3-8B, we selected an example under the **ETS Top (20%)** and **GMTS Top (20%)** settings to illustrate that the output of GMTS is not only more concise but also more accurate compared to ETS:

---

**Question and Ground-Truth**

**Question**: Find the number of ordered pairs $(x, y)$, where both $x$ and $y$ are integers between $-100$ and $100$, inclusive, such that $12x^2 - xy - 6y^2 = 0$.
**Ground-Truth**:

$$\boxed{117}$$

---

**ETS Top (20%)**

Okay, so I need to find the number of ordered pairs $(x, y)$ where both $x$ and $y$ are integers between $-100$ and $100$ inclusive, and they satisfy the equation

$$12x^2 - xy - 6y^2 = 0.$$

Let me think about how to approach this.
First, the equation is quadratic in both $x$ and $y$. Maybe I can factor it or find some relationship between $x$ and $y$. Let me try rearranging the equation. Let me write it again:

$$12x^2 - xy - 6y^2 = 0.$$

Hmm. Maybe I can factor this quadratic equation. Let me see. If I treat this as a quadratic in $x$, then maybe I can factor it or use the quadratic formula. Alternatively, maybe factor it as a quadratic in $y$. Let me try both approaches.
Let me try treating it as a quadratic in $x$. So,

$$12x^2 - yx - 6y^2 = 0.$$

So, in the form $ax^2 + bx + c = 0$, where $a = 12$, $b = -y$, and $c = -6y^2$. Then, using quadratic formula:

$$x = \frac{y \pm \sqrt{y^2 - 4 \cdot 12 \cdot (-6y^2)}}{2 \cdot 12}.$$

Calculating discriminant:

$$D = y^2 - 4 \cdot 12 \cdot (-6y^2) = y^2 + 288y^2 = 289y^2.$$

So

$$\sqrt{D} = \sqrt{289y^2} = 17|y|.$$

Since $y$ is an integer, $|y|$ is non-negative, so $\sqrt{D}$ is $17|y|$.

---

Therefore,
$$x = \frac{y \pm 17|y|}{24}.$$

—

Case 1: $y \geq 0$. Then $|y| = y$. So
$$x = \frac{y \pm 17y}{24}.$$

So two possibilities:
$$x = \frac{y + 17y}{24} = \frac{18y}{24} = \frac{3y}{4},$$
or
$$x = \frac{y - 17y}{24} = \frac{-16y}{24} = -\frac{2y}{3}.$$

Case 2: $y < 0$. Then $|y| = -y$. So
$$x = \frac{y \pm 17(-y)}{24}.$$

Let's compute:
First, with the plus sign:
$$\frac{y + (-17y)}{24} = \frac{-16y}{24} = -\frac{2y}{3}.$$

With the minus sign:
$$\frac{y - (-17y)}{24} = \frac{y + 17y}{24} = \frac{18y}{24} = \frac{3y}{4}.$$

So regardless of the sign of $y$, the solutions for $x$ are
$$x = \frac{3y}{4} \quad \text{or} \quad x = -\frac{2y}{3}.$$

Therefore, for each $y$, $x$ must be either $\frac{3y}{4}$ or $-\frac{2y}{3}$. But since $x$ has to be an integer, this imposes conditions on $y$. Therefore, $3y/4$ must be integer or $-2y/3$ must be integer. Therefore:
Either $4 \mid 3y$ or $3 \mid 2y$.
But since 3 and 4 are coprime, $4 \mid 3y$ implies $4 \mid y$. Similarly, $3 \mid 2y$ implies $3 \mid y$. Therefore, for $x$ to be integer:
Either $y$ is divisible by 4, or $y$ is divisible by 3. Therefore, the solutions are:
- For each $y$ divisible by 4, $x = 3y/4$ is integer. - For each $y$ divisible by 3, $x = -2y/3$ is integer.
If $y$ is divisible by both 3 and 4 (i.e., by 12), then both $x$'s would be integers.

—

Therefore, the total number of solutions would be the number of $y$'s divisible by 4 plus the number of $y$'s divisible by 3. However, we have to be careful not to double count the cases where $y$ is divisible by both 3 and 4, i.e., by 12. Because those $y$'s would be counted in both sets. Therefore, using inclusion-exclusion, total solutions would be:

$$N = (\#\{y : 4 \mid y\}) + (\#\{y : 3 \mid y\}) - (\#\{y : 12 \mid y\}).$$

—

Now we compute.
Range: $-100 \leq y \leq 100$, i.e., 201 integers.
- Number of $y$ divisible by 4: $\lfloor 100/4 \rfloor \times 2 + 1 = 25 \times 2 + 1 = 51$. - Number of $y$ divisible by 3: $\lfloor 100/3 \rfloor \times 2 + 1 = 33 \times 2 + 1 = 67$. - Number of $y$ divisible by 12: $\lfloor 100/12 \rfloor \times 2 + 1 = 8 \times 2 + 1 = 17$.
Thus
$$N = 51 + 67 - 17 = 101.$$

—

Check special cases:
- $y = 0$. Then the equation is $12x^2 = 0 \implies x = 0$. So $(0,0)$ is one solution. This is correctly counted once by the formula.
- $y = 4$. Then $x = 3$. Solution: $(3, 4)$ works.
- $y = 3$. Then $x = -2$. Solution: $(-2, 3)$ works.
- $y = 12$. Then $x = 9$ or $x = -8$, both valid.
All checks out.

—

Finally, also check $x$ range: $-100 \leq x \leq 100$. For $|y| \leq 100$, $x = \frac{3y}{4}$ or $x = -\frac{2y}{3}$ also lies in $[-100, 100]$. Verified.

—

$$\boxed{101}$$

### GMTS Top (20%)

Okay, so I need to find the number of ordered pairs $(x, y)$ where both $x$ and $y$ are integers between $-100$ and 100 inclusive, and they satisfy the equation $12x^2 - xy - 6y^2 = 0$. Hmm, let me think about how to approach this.

First, maybe I can try to factor the equation or simplify it somehow. Let me write down the equation again:

$$12x^2 - xy - 6y^2 = 0.$$

Hmm, quadratic in terms of $x$ and $y$. Maybe I can treat this as a quadratic equation in $x$ or $y$? Let me try treating it as a quadratic in $x$. Let me rearrange the terms:

$$12x^2 - xy - 6y^2 = 0.$$

If I consider this as a quadratic in $x$, then it's of the form $ax^2 + bx + c = 0$, where:

$$a = 12, \quad b = -y, \quad c = -6y^2.$$

Then, using the quadratic formula,

$$x = \frac{-b \pm \sqrt{b^2 - 4ac}}{2a} = \frac{y \pm \sqrt{y^2 - 4 \cdot 12 \cdot (-6y^2)}}{24}.$$

Simplify inside the square root:

$$y^2 - 4 \cdot 12 \cdot (-6y^2) = y^2 + 288y^2 = 289y^2.$$

So

$$\sqrt{289y^2} = 17|y|.$$

Therefore,

$$x = \frac{y \pm 17|y|}{24}.$$

——

Case 1: $y \geq 0$. Then $|y| = y$. Therefore:

$$x = \frac{y \pm 17y}{24}.$$

So two possibilities:

$$x = \frac{18y}{24} = \frac{3y}{4}, \quad \text{or} \quad x = \frac{-16y}{24} = -\frac{2y}{3}.$$

Case 2: $y < 0$. Then $|y| = -y$. Therefore:

$$x = \frac{y \pm 17(-y)}{24}.$$

So:
First possibility: $y + (-17y) = -16y$, hence $x = -\frac{2y}{3}$.
Second possibility: $y - (-17y) = 18y$, hence $x = \frac{3y}{4}$.
——

So regardless of the sign of $y$, the solutions for $x$ are:

$$x = \frac{3y}{4} \quad \text{or} \quad x = -\frac{2y}{3}.$$

Therefore, the solutions to the equation are all pairs $(x, y)$ with $x, y \in [-100, 100]$ such that either $x = 3y/4$ or $x = -2y/3$, with integrality conditions.
——

**Case 1:** $x = 3y/4$.
For $x$ to be integer, $y$ must be divisible by 4. Let $y = 4k$, then $x = 3k$. Range: $-100 \leq 4k \leq 100 \implies -25 \leq k \leq 25$. That gives $25 - (-25) + 1 = 51$ solutions.
**Case 2:** $x = -2y/3$.
For $x$ to be integer, $y$ must be divisible by 3. Let $y = 3m$, then $x = -2m$. Range: $-100 \leq 3m \leq 100 \implies -33 \leq m \leq 33$. That gives $33 - (-33) + 1 = 67$ solutions.
——

**Overlap check:**
Suppose $(x, y)$ satisfies both forms: $(x, y) = (3k, 4k)$ and $(x, y) = (-2m, 3m)$. Then $3k = -2m$ and $4k = 3m$. Solving gives $k = 0, m = 0$, hence $(0, 0)$. So there is exactly one overlap.

**Final count:**

$$51 + 67 - 1 = 117.$$

$$\boxed{117}$$

## A.5 ADDITIONAL SUPPORTING EXPERIMENTS

**The correlation between $E(o_{i,t})$ and $E(o_{i,t}) \cdot \omega_{i,t}$**

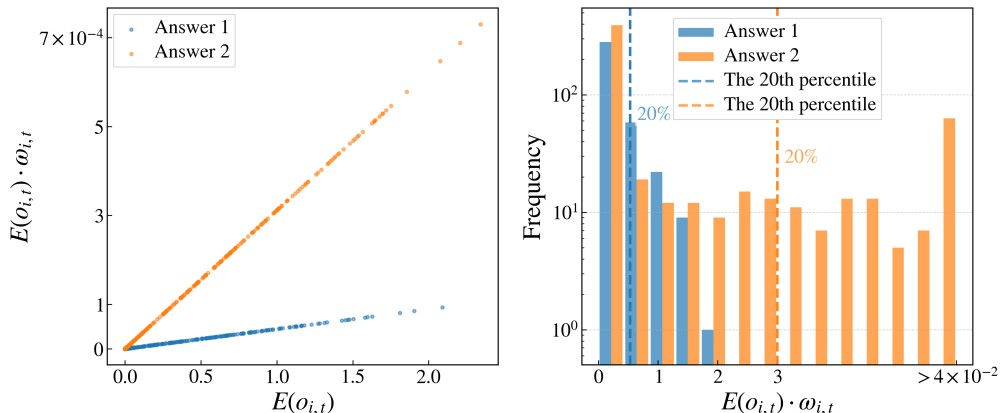

Figure 6: We use DAPO to train Qwen2.5-math-1.5B on MATH-12K dataset under the setting of $G = 16$. At training step 100, we selected two answers with different advantages in one question group. The left figure shows the correlation between $E(o_{i,t})$ and $E(o_{i,t}) \cdot \omega_{i,t}$ while the right figure shows the distributions of the $E(o_{i,t}) \cdot \omega_{i,t}$ together with their 20th percentile boundaries.

## A.6 USE OF LARGE LANGUAGE MODELS

Following the ICLR 2026 policy on LLM usage, we clarify that these tools were used *only* for improving grammar, clarity, and readability of the manuscript. They were not involved in problem formulation, theoretical analysis, algorithm design, implementation, experiments, or interpretation of results. LLMs were also not used for literature search, idea generation, or producing scientific content. All technical work, including proofs, models, and experiments, was conducted solely by the authors.

## A.7 EXAMPLES OF THE TOP 20% TOKENS CHOSEN ON **HIGH** & **LOW** -ADVANTAGE ANSWERS BY **ETS** AND **GMTS**

In this section, we use DAPO to train Qwen2.5-math-1.5B on the MATH-12K dataset under the setting $G = 16$. At training step 100, we examine how ETS and GMTS select the top 20% tokens on low-advantage and high-advantage answers, and provide concrete examples. Specifically, the selected tokens are highlighted in red in the figures.

As shown there, GMTS produces a more balanced and conservative selection. This illustrates that GMTS does not blindly favor high-entropy tokens, but also adapts its selection according to the advantage value, leading to a more stable and moderated behavior.

Examples of the Top 20% Tokens Chosen on **high-Advantage** answers by **ETS**:

| The | number | of | diagon | als | \( | D | \) | in | a |
|---|---|---|---|---|---|---|---|---|---|
| polygon | with | \( | n | \) | sides | is | given | by | the |
| formula | : | \[ | D | = | \ | frac | {n | (n | - |
| 3 | )} | { | 2 | } | \ | ] | The | shape | we |
| 're | dealing | with | here | is | a | dec | agon | , | which |
| has | \( | n | = | | 1 | 0 | \) | sides | . |
| We | will | substitute | \( | n | = | | 1 | 0 | \) |
| into | the | formula | to | find | the | number | of | diagon | als |
| in | a | convex | dec | agon | . | Let | 's | calculate | this |
| using | Python | to | ensure | accuracy | : | ``` | python | | # |
| Number | of | sides | in | a | dec | agon | | n | = |
| | 1 | 0 | | # | Formula | to | calculate | diagon | als |
| in | a | polygon | with | n | sides | | D | = | ( |
| n | * | ( | n | - | | 3 | )) | / | |
| 2 | | print | (D | ) | `` | ` | ``` | output | |
| 3 | 5 | . | 0 | | `` | ` | According | to | our |
| calculation | , | a | convex | dec | agon | has | \ | (\ | boxed |
| { | 3 | 5 | }\ | ) | diagon | als | . | | |

Examples of the Top 20% Tokens Chosen on **high-Advantage** Answers by **GMTS**:

| The | number | of | diagon | als | \( | D | \) | in | a |
|---|---|---|---|---|---|---|---|---|---|
| polygon | with | \( | n | \) | sides | is | given | by | the |
| formula | : | \[ | D | = | \ | frac | {n | (n | - |
| 3 | )} | { | 2 | } | \ | ] | The | shape | we |
| 're | dealing | with | here | is | a | dec | agon | , | which |
| has | \( | n | = | | 1 | 0 | \) | sides | . |
| We | will | substitute | \( | n | = | | 1 | 0 | \) |
| into | the | formula | to | find | the | number | of | diagon | als |
| in | a | convex | dec | agon | . | Let | 's | calculate | this |
| using | Python | to | ensure | accuracy | : | ``` | python | | # |
| Number | of | sides | in | a | dec | agon | | n | = |
| | 1 | 0 | | # | Formula | to | calculate | diagon | als |
| in | a | polygon | with | n | sides | | D | = | ( |
| n | * | ( | n | - | | 3 | )) | / | |
| 2 | | print | (D | ) | `` | ` | ``` | output | |
| 3 | 5 | . | 0 | | `` | ` | According | to | our |
| calculation | , | a | convex | dec | agon | has | \ | (\ | boxed |
| { | 3 | 5 | }\ | ) | diagon | als | . | | |

Examples of the Top 20% Tokens Chosen on **low-Advantage** Answers by **ETS**:

| | | | | | | | | | |
|---|---|---|---|---|---|---|---|---|---|
| Given | the | quadratic | equation | \( | x | ^ | 2 | + | |
| 5 | x | + | | 7 | = | | 0 | \ | ), |
| we | need | to | compute | the | value | of | \ | (( | r |
| - | | 1 | )( | r | + | | 2 | )( | r |
| + | | 6 | )( | r | + | | 3 | )\ | ), |
| where | \( | r | \) | is | one | of | the | roots | of |
| the | equation | . | Let | 's | break | it | down | into | steps |
| : | 1 | . | ** | Find | the | roots | of | the | quadratic |
| equation | \( | x | ^ | 2 | + | | 5 | x | + |
| | 7 | = | | 0 | \ | )** | : | | - |
| The | roots | of | the | quadratic | equation | \( | ax | ^ | 2 |
| + | bx | + | c | = | | 0 | \) | are | given |
| by | the | quadratic | formula | \( | r | = | \ | frac | {- |
| b | \ | pm | \ | sqrt | { | b | ^ | 2 | - |
| | 4 | ac | }} | { | 2 | a | }\ | ). | 2 |
| . | ** | Compute | the | expression | \ | (( | r | - | |
| 1 | )( | r | + | | 2 | )( | r | + | |
| 6 | )( | r | + | | 3 | )\ | )** | : | |
| - | Substitute | the | roots | back | into | the | expression | and | simplify |
| . | Let | 's | implement | this | in | Python | to | compute | the |
| final | answer | . | ``` | python | | import | symp | y | as |
| sp | | # | Define | the | quadratic | equation | | x | = |
| sp | .s | ymbols | (' | x | ') | qu | adratic | _eq | = |
| x | ** | 2 | + | | 5 | *x | + | | 7 |
| | # | Find | the | roots | of | the | quadratic | equation | |
| roots | = | sp | .solve | (qu | adratic | _eq | , | x | ) |
| r | = | roots | [ | 0 | ] | | # | Let | 's |
| choose | one | of | the | roots | | # | Define | the | expression |
| ( | r | - | | 1 | )( | r | + | | 2 |

| )( | r | + | | 6 | )( | r | + | | 3 |
|---|---|---|---|---|---|---|---|---|---|
| ) | expression | = | ( | r | - | | 1 | )*( | r |
| + | | 2 | )*( | r | + | | 6 | )*( | r |
| + | | 3 | ) | # | Compute | the | value | of | the |
| expression | | result | = | expression | .eval | f | () | print | (result |
| ) | `` | ` | ``` | output | | 8 | 4 | . | 0 |
| 0 | 0 | 0 | 0 | 0 | 0 | 0 | 0 | 0 | 0 |
| 0 | 0 | | `` | ` | The | value | of | the | expression |
| \ | (( | r | - | | 1 | )( | r | + | |
| 2 | )( | r | + | | 6 | )( | r | + | |
| 3 | )\ | ), | where | \( | r | \) | is | a | root |
| of | the | quadratic | equation | \( | x | ^ | 2 | + | |
| 5 | x | + | | 7 | = | | 0 | \ | ), |
| is | \ | (\ | boxed | { | 8 | 4 | }\ | ). | |

Examples of the Top 20% Tokens Chosen on low-Advantage Answers by **GMTS**:

| | | | | | | | | | |
|---|---|---|---|---|---|---|---|---|---|
| Given | the | quadratic | equation | \( | x | ^ | 2 | + | |
| 5 | x | + | | 7 | = | | 0 | \ | ), |
| we | need | to | compute | the | value | of | \ | (( | r |
| - | | 1 | )( | r | + | | 2 | )( | r |
| + | | 6 | )( | r | + | | 3 | )\ | ), |
| where | \( | r | \) | is | one | of | the | roots | of |
| the | equation | . | Let | 's | break | it | down | into | steps |
| : | 1 | . | ** | Find | the | roots | of | the | quadratic |
| equation | \( | x | ^ | 2 | + | | 5 | x | + |
| | 7 | = | | 0 | \ | )** | : | | - |
| The | roots | of | the | quadratic | equation | \( | ax | ^ | 2 |
| + | bx | + | c | = | | 0 | \) | are | given |
| by | the | quadratic | formula | \( | r | = | \ | frac | {- |
| b | \ | pm | \ | sqrt | { | b | ^ | 2 | - |
| | 4 | ac | }} | { | 2 | a | }\ | ). | 2 |
| . | ** | Compute | the | expression | \ | (( | r | - | |
| 1 | )( | r | + | | 2 | )( | r | + | |
| 6 | )( | r | + | | 3 | )\ | )** | : | |
| - | Substitute | the | roots | back | into | the | expression | and | simplify |
| . | Let | 's | implement | this | in | Python | to | compute | the |
| final | answer | . | ``` | python | | import | symp | y | as |
| sp | | # | Define | the | quadratic | equation | | x | = |
| sp | .s | ymbols | (' | x | ') | qu | adratic | _eq | = |
| x | ** | 2 | + | | 5 | *x | + | | 7 |
| | # | Find | the | roots | of | the | quadratic | equation | |
| roots | = | sp | .solve | (qu | adratic | _eq | , | x | ) |
| r | = | roots | [ | 0 | ] | | # | Let | 's |
| choose | one | of | the | roots | | # | Define | the | expression |
| ( | r | - | | 1 | )( | r | + | | 2 |

| | | | | | | | | | |
|---|---|---|---|---|---|---|---|---|---|
| )( | r | + | | 6 | )( | r | + | | 3 |
| ) | expression | = | ( | r | - | | 1 | )*( | r |
| + | | 2 | )*( | r | + | | 6 | )*( | r |
| + | | 3 | ) | # | Compute | the | value | of | the |
| expression | | result | = | expression | .eval | f | () | print | (result |
| ) | `` | ` | ``` | output | | **8** | **4** | . | 0 |
| 0 | 0 | 0 | 0 | 0 | 0 | 0 | 0 | 0 | 0 |
| 0 | 0 | | `` | ` | The | value | of | the | expression |
| \ | (( | r | - | | 1 | )( | r | + | |
| 2 | )( | r | + | | 6 | )( | r | + | |
| 3 | )\ | ), | where | \( | r | \) | is | a | root |
| of | the | quadratic | equation | \( | x | ^ | 2 | + | |
| 5 | x | + | | 7 | = | | 0 | \ | ), |
| is | \ | (\ | boxed | { | 8 | 4 | }\ | ). | |

