# OpenReview forum: "GMTS: Gradient Magnitude-based Token Selection Improves RLVR Training for LLM Reasoning"
_ICLR.cc/2026/Conference — Submitted to ICLR 2026_

### Official Review · Reviewer_Q1A3 · 2025-10-30

**Soundness:** 3
**Presentation:** 3
**Contribution:** 2
**Rating:** 4
**Confidence:** 4

**Summary:**

The paper builds up on prior work showing that during finetuning with RL, training only on the tokens with highest entropy improves performance. It is found that this relates to the gradient of the tokens, and actually selecting tokens with largest gradient improves performance even more. Experiments with GRPO and DAPO finetuning on math benchmarks demonstrate that the proposed method, GMTS, improves over entropy token selection (ETS) by 1-2%.

**Strengths:**

1. The paper presents a convincing analysis relating entropy to gradients and derives a new method with performance gains.
2. There are experiments on several math benchmarks, various training methods and models.

**Weaknesses:**

1. The differences between using the proposed method, GMTS, and the base model, are very small, usually <2% accuracy. The ETS paper showed that the differences become more pronounced when using larger models, e.g. 16B or 32B, but I assume this is computationally expensive. If possible, it would be good to show a comparison to ETS on a larger model.
2. Some results are not really supporting the value of sub-set selection, e.g. in Figure 5 the curve is pretty much flat (selecting 10% or 90% best tokens has similar performance), and in Figure 3a, it looks like ETS might be better when just using more training steps.
3. In the end, the GMTS seems to be a simple weighting of the entropy by \omega. How different are those two usually? It would be good to see a scatterplot of  E(o_i) vs E(o_i) * \omega_i.

**Questions:**

1. Was the model trained until convergence (figure 3a)?
2. Are there other advantages of GMTS, e.g. more conscise answers, apart from the 1% performance gain?
3. How is the relation in Equation 3 derived? A first order Taylor expansion around x_0=1 would just yield (1-p_k) for me and not p_k (1-p_k).

---

> ### Author Response · Authors · 2025-11-27
> **Part 1/2**
>
> We thank the reviewer for the positive feedback on our motivation, analysis, and the breadth of experiments, and for the thoughtful comments and questions. Below, we address the concerns in detail.
>
> > **Q1:** The differences between using the proposed method, GMTS, and the base model are very small, ... on a larger model.
>
>  **A1:**
> We fully agree with the reviewer's point about evaluating GMTS on larger and stronger models. Unfortunately, due to limited computational resources, we have no time to run full RL training on very large models (e.g., 16B or 32B). To facilitate further verification, we have provided an anonymized code repository link in the revised manuscript so that stronger models can be tested by others. Within our computational budget, we conducted the following additional experiments to better assess the generality of GMTS, which are shown in **Summary Part 2, 3, and 4**.
>
> > **Q2:** Some results are not really supporting the value of sub-set selection, e.g. in Figure 5, the curve is pretty much flat (selecting 10% or 90% best tokens has similar performance), and in Figure 3a, it looks like ETS might be better when just using more training steps.
>
>  **A2:**
>  We thank the reviewer for the opportunity to clarify these points.
>
> (1) **For sub-set selection**
> The reason that selecting 90% of the tokens yields similar performance to selecting only 10% is twofold: when using 90% of the tokens, many low gradient tokens are included and can introduce noise or even negatively affect optimization at the trajectory level, while using only 10% of the tokens does not provide a sufficient learning signal. This suggests that choosing an appropriate selection ratio is crucial for balancing information content and noise in token selection.
>
> (2) **For testing curves**
> For **Figure 3(a)**, the plotted curve is not the raw training curve, but the test accuracy curve smoothed by a rolling window over validation evaluations. Because we evaluate on the test/validation set quite frequently during RL training, this smoothing is applied only to make the curve readable; it does not indicate that the model has not converged. In terms of the actual training dynamics, the training curves for both ETS and GMTS have already converged under our compute budget.
>
> > **Q3:** In the end, the GMTS seems to be a simple weighting of the entropy by \omega. How different are those two usually? It would be good to see a scatterplot of E(o_i) vs E(o_i) * \omega_i.
>
>  **A3:**
> We are happy to provide additional illustrative scatter plots to demonstrate this phenomenon. These have been added in **Appendix A.5** of the revised manuscript, and we kindly refer the reviewer to that section for details.

---

> > ### Author Response · Authors · 2025-11-27
> > **Part 2/2**
> >
> > > **Q4:** Was the model trained until convergence (figure 3a)? Are there other advantages of GMTS, e.g. more concise answers, apart from the 1% performance gain?
> >
> >  **A4:**
> > We thank the reviewer for these insightful questions.
> > (1) On convergence (Figure 3(a)).
> > In terms of training dynamics, the models are indeed trained until convergence under our compute budget.
> >
> > (2) On advantages beyond the ~1% performance gain.
> > Beyond the numerical accuracy improvements, we observe two additional advantages of GMTS:
> >
> > **1. Generalization to other reasoning tasks.**
> >
> >  In our supplementary experiments, we apply GMTS to a different reasoning benchmark, **commonsense_qa**, which the details are in **Summary Part 2 and 3**. Under the same evaluation protocol, GMTS still outperforms ETS, indicating that its benefit is not limited to math-only datasets but also transfers to other reasoning tasks.
> >
> > **2. More concise and cleaner answers.**
> >
> > We also examined the generation quality and found that GMTS tends to produce more concise and less cluttered solutions than ETS. Concretely, the average answer length under GMTS is noticeably shorter while maintaining or improving accuracy, suggesting that GMTS reduces unnecessary verbosity in the reasoning traces. We have put a table below that reports the **avg@16** output lengths for ETS vs. GMTS.
> >
> > | Qwen2.5-1.5b| AIME2024 | AMC23| MATH-500  |
> > |-|-|-|- |
> > | DAPO | 806.4| 629.2 | 677.9|
> > | +ETS Top (20%)| 794.4 | 611.4 | 593.2|
> > | **+GMTS Top (20%)** | **760.3** | **544.2** | **580.1** |
> >
> > |Qwen2.5-7b | AIME2024|AMC23|MATH-500|
> > |-|-|-|-|
> > | DAPO|904.2 | 704.2| 636.8|
> > | +ETS Top (20%) | 877.5 | 692.4|564.2|
> > | **+GMTS Top (20%)** | **820.1** | **662.7** | **544.2** |
> >
> > > **Q5:** How is the relation in Equation 3 derived? A first order Taylor expansion around x_0=1 would just yield (1-p_k) for me and not p_k (1-p_k).
> >
> >  **A5:**
> > Thank you for your insightful comment and for pointing out that our original description was incorrect description. In the revised manuscript, we have corrected this wording. We have clarified this point in the revision by explicitly stating that: **Equation (3)** should be viewed as a **Gini-style** approximation, which in other fields is sometimes used as a simple approximation or surrogate for entropy [1] [2]. Our intention here was not to claim a strict equivalence or to provide a rigorous proof of a particular identity, but rather to offer an insight into why entropy and gradient magnitude can exhibit an approximately linear correlation. This clarification does not affect our main contributions, you can refer to **Summary Part 1** for our contributions. GMTS is motivated by the correlation between entropy and gradient magnitude, and GMTS itself does not rely on any strict equality between entropy and the Gini index.
> >
> > We are very grateful to the reviewer for the constructive suggestions and for pointing out the parts that were not sufficiently rigorous. We hope that the revisions in our updated manuscript will address these concerns and dispel any remaining doubts.
> >
> > [1] Tatti N. Approximating splits for decision trees quickly in sparse data streams[C]//Proceedings of the 2025 SIAM International Conference on Data Mining (SDM). Society for Industrial and Applied Mathematics, 2025: 647-655.
> >
> > [2] Martino L, Elvira V. Effective sample size approximations as entropy measures[J]. Computational Statistics, 2025: 1-32.

---

### Official Review · Reviewer_S8Hs · 2025-10-30

**Soundness:** 2
**Presentation:** 3
**Contribution:** 2
**Rating:** 4
**Confidence:** 4

**Summary:**

This paper introduces Gradient Magnitude-based Token Selection (GMTS), a novel method for improving the training of Large Language Models (LLMs) on reasoning tasks using Reinforcement Learning with Verifiable Rewards (RLVR). The authors first establish a relationship between a token's entropy and its log-probability gradient magnitude, providing an explanation for the success of prior work on Entropy-based Token Selection (ETS). However, they argue that entropy alone is an insufficient proxy for token importance across different generated answers, as it does not account for variations in answer-level rewards. To address this, GMTS proposes a new metric for token importance that approximates the gradient magnitude by combining token entropy with the answer-level advantage signal. The paper demonstrates through extensive experiments on mathematical reasoning benchmarks that training on the top 20% of tokens ranked by GMTS consistently and significantly outperforms the ETS baseline across various model sizes (1.5B, 7B, 8B) and RLVR algorithms (DAPO, GRPO).

**Strengths:**

1. The paper provides a very clear and intuitive motivation. It begins by explaining why entropy-based selection (ETS) works, the correlation between entropy and gradient magnitude within a single answer, and then clearly demonstrates its limitations. The analysis showing that gradient distributions shift for answers with different advantages (Figure 2) provides a compelling argument that entropy alone is not a complete picture, motivating the need for a more robust metric like GMTS.
2. The proposed GMTS method is simple to understand and implement. It leverages quantities (advantage, entropy) that are already available during standard RLVR training, thus introducing minimal computational overhead. Despite its simplicity, the method is shown to be highly effective, yielding consistent and often significant performance improvements over a strong baseline (ETS) on a variety of challenging math reasoning benchmarks.

**Weaknesses:**

1. The performance gains of GMTS over ETS and DAPO, while consistent, appear to be in the range of 1-2 percentage points. This marginal improvement may not be sufficient to demonstrate the method's broad effectiveness and generality. Additionally, since the method is a modification of ETS, the overall contribution might seem relatively marginal.

2.	Figure 3(b) shows that entropy still trends downwards, suggesting a collapse phenomenon. This indicates the method may not fully mitigate issues in long training runs (e.g., thousands of steps). When overall entropy drops to a low point, the entropies of the top 20% of tokens will also be very low and difficult to distinguish. Can GMTS continue to provide gains in such a scenario?

**Questions:**

1.	Could you provide a comparison of GMTS against ETS and DAPO on models known for long chain-of-thought reasoning, such as DeepSeek-Qwen2.5-1.5b-Distill and DeepSeek-Qwen2.5-7b-Distill?
2.	Was the RL training conducted exclusively on the Math-12k dataset? Given that this dataset might be relatively simple, have you considered evaluating your method using more challenging or diverse training data, such as the datasets from Deep-Scaler, DAPO, or Skywork-OR1?
3.	Could you provide a visualization that illustrates which specific tokens are selected by ETS versus GMTS during training on a given example?

---

> ### Author Response · Authors · 2025-11-27
>
> We sincerely thank the reviewer for the positive assessment of our motivation and the advantages of GMTS. We have carefully read and reflected on your comments and are very grateful for the valuable suggestions. We hope that our responses below can adequately address your concerns.
>
> > **Q1:** The performance gains of GMTS over ETS and DAPO, while consistent, appear to be in the range of 1-2 percentage points. This marginal improvement may not be sufficient to demonstrate the method's broad effectiveness and generality. Additionally, since the method is a modification of ETS, the overall contribution might seem relatively marginal.
>
> **A1:**
> We understand the reviewer's concern about the performance and the contribution of our work. We have explained it in the **Summary Part 4**. Please see this part for more details.
>
> > **Q2:** Figure 3(b) shows that entropy still trends downwards, suggesting a collapse phenomenon. This indicates the method may not fully mitigate issues in long training runs (e.g., thousands of steps). When overall entropy drops to a low point, the entropies of the top 20% of tokens will also be very low and difficult to distinguish. Can GMTS continue to provide gains in such a scenario?
>
> **A2:**
> We sincerely thank the reviewer for this valuable suggestion. In the current work, we do not specifically analyze or address the phenomenon of entropy collapse. Our focus is on studying the relationship between entropy and gradient magnitude, rather than on the dynamics of entropy collapse itself. However, prior work such as [1] [2] has investigated entropy collapse in RL-style training. Since GMTS is implemented as a plug-and-play example-weighting scheme and can be applied in parallel with entropy-based regularization and collapse-mitigation methods, in principle, they can be used in combination. We leave a systematic study of such combinations to future work.
>
> > **Q3:** Could you provide a comparison of GMTS against ETS and DAPO on models known for long chain-of-thought reasoning, such as DeepSeek-Qwen2.5-1.5b-Distill and DeepSeek-Qwen2.5-7b-Distill?
>
> **A3:**
> We sincerely thank the reviewer for the valuable suggestions regarding additional experiments. We have fully utilized our available computational resources and conducted further experiments in **Summary Part 2, and 3**.
>
> > **Q4:** Was the RL training conducted exclusively on the Math-12k dataset? Given that this dataset might be relatively simple, have you considered evaluating your method using more challenging or diverse training data, such as the datasets from Deep-Scaler, DAPO, or Skywork-OR1?
>
> **A4:**
> Thank you for this insightful question. Our RL training is not restricted to Math-12k. For the **Qwen3-8B** model, we already use **DAPO-MATH** as the training dataset, which is more challenging than Math-12k and specifically designed for mathematical reasoning. For the other backbones, the main RL experiments are based on Math-12k.
>
> As mentioned above, in our newly added experiments, we further move beyond **Math-12k and DAPO-MATH** to test GMTS on more generalized data and tasks. The details are in **Summary Part 2 and 3**. Across these new settings, the results consistently show that GMTS outperforms ETS under the same evaluation protocol. This demonstrates that the advantage of GMTS is not limited to a single dataset, but extends to more challenging math corpora and even to a different reasoning domain.
>
> > **Q5:** Could you provide a visualization that illustrates which specific tokens are selected by ETS versus GMTS during training on a given example?
>
>  **A5:**
> We are pleased to provide this example. In the revised manuscript, **Appendix A.7** presents a case study where we compare the token selections of ETS and GMTS under both **low-advantage** and **high-advantage** conditions, showing a clear difference between the two methods.
>
> We sincerely thank the Reviewer for the constructive suggestions that helped improve the quality of our paper, and we hope that our responses satisfactorily address your concerns.
>
> [1] Hao Z, Wang H, Liu H, et al. Rethinking entropy interventions in RLVR: An entropy change perspective[J]. arXiv preprint arXiv:2510.10150, 2025.
>
> [2] Wang C, Li Z, Bai J, et al. Arbitrary Entropy Policy Optimization: Entropy Is Controllable in Reinforcement Finetuning[J]. arXiv preprint arXiv:2510.08141, 2025.

---

### Official Review · Reviewer_P374 · 2025-10-31

**Soundness:** 3
**Presentation:** 3
**Contribution:** 3
**Rating:** 6
**Confidence:** 3

**Summary:**

This paper proposes GMTS (Gradient Magnitude-based Token Selection), a technique designed to enhance Reinforcement Learning with Verifiable Rewards (RLVR) for reasoning tasks in large language models.
Building on prior work (ETS), which improved reasoning by training only on high-entropy tokens, the authors identify a limitation: entropy correlates with gradient magnitude within a single answer but not across answers with different reward values.
GMTS addresses this by weighting token importance using the product of entropy and the gradient coefficient derived from RLVR advantage terms.
Experiments on multiple Qwen math models (1.5B, 7B, and 8B) across six reasoning benchmarks demonstrate consistent performance improvements with minimal computational overhead.

**Strengths:**

[S1] The authors provide a clear and well-motivated problem formulation, and the overall argument for improving token selection in RLVR training is convincing.

[S2] The presentation is strong: the paper is clearly written, well-organized, and effectively communicates both the motivation and methodology.

**Weaknesses:**

[W1] Comparison with ETS. The paper’s central motivation that entropy-based token selection (ETS) fails because entropy varies across samples with different rewards is only qualitatively supported. While the authors provide correlation analyses between entropy and gradient magnitude, no concrete examples are shown where ETS explicitly selects misleading or low-reward tokens. As a result, the argument that “entropy cannot reliably measure token importance across answers” remains intuitive rather than empirically verified. A few explicit case studies or quantitative analyses (e.g., proportion of high-entropy tokens in low-reward trajectories) would significantly strengthen the paper’s motivation and clarity.

[W2] Variation of gradients. The paper claims that GMTS can be applied to both GRPO and DAPO training, but it does not discuss how the absence of a KL regularization term in DAPO affects the underlying gradient structure. The theoretical motivation of GMTS, that token importance correlates with gradient magnitude, is derived assuming a KL-regularized objective (as in GRPO), where the gradient is partially stabilized by the KL term. In contrast, DAPO omits this regularization, leading to higher variance and potentially different gradient scaling across tokens. The paper does not analyze whether GMTS’s gradient-based weighting remains theoretically valid or empirically stable under this setting.

[W3] Generalization to other domains. The experiments focus exclusively on mathematical reasoning benchmarks, such as AIME, AMC, and Minerva, which limits the generality of the conclusions. Since GMTS is proposed as a general token selection method for RLVR training, its effectiveness on other domains (e.g., commonsense reasoning, code generation, or science) remains unverified. It is unclear whether the observed improvements are specific to math reasoning, where token uncertainty and reward structures are relatively well-behaved, or if they extend to more diverse or noisy tasks.

**Questions:**

- Could the authors provide specific examples or quantitative evidence showing cases where ETS selects high-entropy tokens from low-reward answers, illustrating how it fails across samples? (W1)

- As DAPO lacks the KL regularization term, why is GMTS expected to remain generally applicable across GRPO variants with different objective forms? In other words, what makes the method robust to whether or not the training objective includes KL regularization? (W2)

- Does GMTS demonstrate generality beyond mathematical reasoning tasks, and is it expected to perform similarly in other domains? (W3)

---

> ### Author Response · Authors · 2025-11-27
>
> We are very grateful for the reviewer's positive evaluation and for the recognition of GMTS. We have carefully read and considered your suggestions. Our responses are provided below.
>
> >  **Q1:** Could the authors provide specific examples or quantitative evidence showing cases where ETS selects high-entropy tokens from low-reward answers, illustrating how it fails across samples? (W1)
>
> **A1:**
> In **Appendix A.7** of the revised manuscript, we provide a qualitative example comparing the token selections of ETS and GMTS under both high-advantage and low-advantage conditions. As shown there, GMTS produces a more balanced and conservative selection. This illustrates that GMTS does not blindly favour high-entropy tokens, but also adapts its selection according to the advantage value, leading to a more stable and moderated behaviour.
>
> >  **Q2:** As DAPO lacks the KL regularization term, why is GMTS expected to remain generally applicable across GRPO variants with different objective forms? In other words, what makes the method robust to whether or not the training objective includes KL regularization? (W2)
>
> **A2:**
> Thanks for your insightful question. The robustness of GMTS remains consistent regardless of the specific RLVR objective function.
> As derived in the manuscripts (**lines 195-197**). The gradient of a token $o_{i,t}$ in GRPO framework can be is:
>
> $$
> \nabla\_\theta \ell\_{i,t}(\theta) =  \underbrace{\left(
> r\_{i, t}(\theta) A\_{i,t}\cdot \mathbb{I}\_{\epsilon\_1, \epsilon\_2}(r\_{i,t}(\theta),A\_{i,t})+\beta \frac{\pi\_{\text{ref}}(o\_{i,t}|\boldsymbol{q},\boldsymbol{o}\_{i,<t})}{\pi\_{\theta}(o\_{i,t}|\boldsymbol{q}, \boldsymbol{o}\_{i,<t})}-\beta\right)}\_{=:\ \omega\_{i,t}(\theta)}\nabla\_\theta \log \pi\_\theta (o\_{i,t} |\boldsymbol{q},\boldsymbol{o}\_{i,<t})
> $$
>
> The GMTS metric, $\delta\_{i,t}(\theta) = |E(o\_{i,t})\cdot\omega\_{i,t}(\theta)|$ consists of two parts:
>
> 1. The coefficient $\omega_{i,t}(\theta)$, which depends on the specific objective (**e.g., $\beta=0$ in DAPO**)
> 2. The correlation between $||\nabla\_\theta \log \pi\_\theta (o\_{i,t} |\boldsymbol{q},\boldsymbol{o}\_{i,<t})||$ and token entropy $E(o\_{i,t})$ as shown in Figure 2. Crucially, the relationship highlighted in Figure 2—the approximate linear correlation between the log-probability gradient and entropy—is an intrinsic property of the language model's output distribution. It is not dependent on the presence or absence of the KL penalty in the RL objective.
>
> Therefore, while  $\omega\_{i,t}(\theta)$ varies with the objective, the core mechanism by which GMTS identifies important tokens via the entropy-magnitude relationship remains valid and effective across different RLVR algorithm variants, making the method broadly applicable.
>
> > **Q3:** Does GMTS demonstrate generality beyond mathematical reasoning tasks, and is it expected to perform similarly in other domains? (W3)
>
> **A3:**
> We are very glad that you raised this point, and we also appreciate the opportunity to demonstrate the advantages of GMTS on other datasets. In our additional experiments, we extend GMTS to **commonsense_qa**, a common-sense reasoning benchmark, and train it on both **Qwen2.5-1.5B** and **Qwen2.5-7B**. As shown in  **Summary Part 2 and 3**, GMTS continues to maintain a clear advantage over the baselines in these settings.
>
> We sincerely appreciate your insightful comments and hope that our responses have addressed your concerns.

---

### Official Review · Reviewer_DEn5 · 2025-11-01

**Soundness:** 2
**Presentation:** 2
**Contribution:** 2
**Rating:** 4
**Confidence:** 4

**Summary:**

This paper proposes Gradient Magnitude-based Token Selection (GMTS), a method to improve reinforcement learning with verifiable rewards (RLVR) for training large language models (LLMs) in reasoning tasks. While prior work emphasized high-entropy tokens for training, the authors find that entropy alone fails to consistently reflect token importance due to variations in answer-level reward signals. GMTS addresses this by ranking tokens based on their gradient magnitude, which better captures their contribution to learning. Experiments on mathematical reasoning benchmarks show GMTS outperforms entropy-based selection, achieving notable performance gains (+1.55 to +1.85 across Qwen models).

**Strengths:**

The paper focuses on improving the performance of reinforcement learning (RL) for large language models (LLMs), which is a critical and impactful area of research. RL plays a key role in enhancing LLM reasoning capabilities, and exploring methods to optimize RL training is relevant to advancing the field.

**Weaknesses:**

1. The paper does not provide code or implementation details, making it difficult for researchers to replicate the results or apply GMTS to other models. Providing well-documented code would enhance the paper's impact and accessibility.

2. While the experiments use multiple backbone models (Qwen2.5-math-1.5B, Qwen2.5-math-7B, and Qwen3-8B), these are relatively small-scale models. The effectiveness of GMTS on larger models (e.g., 13B, 32B) remains untested. Expanding experiments to larger-scale LLMs would strengthen the paper's claims and demonstrate broader applicability.

3. The reported baseline performance on benchmarks like Math500 and AIME24 is significantly lower than what is typically reported in official documentation or other math-related papers. This discrepancy raises questions about the experimental setup or evaluation methodology. The authors should clarify how these results were obtained and ensure alignment with standard practices to validate their findings.

**Questions:**

refer to Weaknesses

---

> ### Author Response · Authors · 2025-11-27
>
> We thank the Reviewer for the constructive comments and for the careful evaluation of our manuscript. We have carefully considered the concerns and issues raised in the review. Below, we respond to each of the points raised by the Reviewer in detail, and we hope that our clarifications will address your concerns.
>
> > **Q1:** The paper does not provide code or implementation details, making it difficult for researchers to replicate the results or apply GMTS to other models. Providing well-documented code would enhance the paper's impact and accessibility.
>
> **A1:**
> We fully agree with the reviewer on the importance of reproducibility. In the revised manuscript, we have **provided an anonymized GitHub link** to our code repository, which includes the implementations of both versions of GMTS as well as all corresponding configuration files. The detailed experimental setups and hyperparameter configurations are described in the main experimental section **4.1** and further documented in the **Appendix A.3** for easy reference.
>
> > **Q2:** While the experiments use multiple backbone models (Qwen2.5-math-1.5B, Qwen2.5-math-7B, and Qwen3-8B), these are relatively small-scale models. The effectiveness of GMTS on larger models (e.g., 13B, 32B) remains untested. Expanding experiments to larger-scale LLMs would strengthen the paper's claims and demonstrate broader applicability.
>
> **A2:**
> We thank the reviewer for proposing this problem. However, due to the limited computational resources, we do not have enough time to train substantially larger-scale models. To strengthen the empirical evidence for GMTS from other perspectives under this constraint, we conducted two additional sets of experiments. Please refer to the **Summary Part 2, and 3** for details.
>
> > **Q3:** The reported baseline performance on benchmarks like Math500 and AIME24 is significantly lower than what is typically reported in official documentation or other math-related papers. This discrepancy raises questions about the experimental setup or evaluation methodology. The authors should clarify how these results were obtained and ensure alignment with standard practices to validate their findings.
>
> **A3:**
> Thanks for your comments. Here are our setting details for evaluation:
>
> (1) For **Qwen2.5-math-1.5B** and **Qwen2.5-math-7B**, we set the maximum generation length to **2048** tokens, and for **Qwen3-8B** we set it to **4096** tokens. and all models share the same decoding hyperparameters,with **top-p = 1.0** and **temperature = 1.0**.
>
> (2) Instead of reporting **pass@1**, we report **avg@16**, which averages performance over **16** independent decoding runs. This metric typically yields lower absolute accuracy than **pass@1**, but it provides a stricter and more informative measure of the stability and robustness of model responses.
>
> Compared with Qwen2.5 math technical report [1], they do not provide the performance without a few shots on the Qwen2.5-math base model. Compared with Qwen3's technical report [2], we have similar performance on MATH500.  If you have any further concerns or questions, we would be very glad to address them.
>
> We sincerely thank the Reviewer for the insightful and constructive feedback. We hope that these revisions and clarifications address your concerns and strengthen the contribution and reliability of our work.
>
> [1] Yang A, Zhang B, Hui B, et al. Qwen2. 5-math technical report: Toward mathematical expert model via self-improvement[J]. arXiv preprint arXiv:2409.12122, 2024.
>
> [2] Yang A, Li A, Yang B, et al. Qwen3 technical report[J]. arXiv preprint arXiv:2505.09388, 2025.

---

### Official Review · Reviewer_X1D1 · 2025-11-03

**Soundness:** 3
**Presentation:** 3
**Contribution:** 3
**Rating:** 4
**Confidence:** 4

**Summary:**

This paper proposes a Gradient Magnitude-based Token Selection (GMTS) method aimed at improving the training efficiency of large language models in mathematical reasoning tasks under reinforcement learning with verifiable rewards. The core idea of this method is to more accurately identify subsets of tokens that are critical for training by analyzing their gradient magnitudes rather than relying on traditional entropy-based measures. The approach incorporates an importance scoring mechanism that integrates entropy and advantage signals, and during training, parameter updates are performed only on high-scoring tokens. Experiments demonstrate that GMTS enhances model performance compared to other methods across mathematical reasoning benchmarks.

**Strengths:**

- The paper proposes using gradient magnitude as a more stable and robust importance metric. The analysis of the relationship between entropy and gradient magnitude provides a theoretical foundation for the method.

- GMTS is designed as a plug-and-play module that can be easily incorporated into existing RLVR frameworks, which enhances its practical value and impact.

**Weaknesses:**

- The idea of using gradients to identify important words is rather common and has been explored in various previous studies.

- The evaluation is restricted to models under 8B parameters, making it unclear whether the proposed GMTS can generalize to larger and more capable LLMs.

- The experimental gains reported in Tables 1 and 2 are modest, raising concerns about the practical impact of the proposed method.

- The experiments primarily focus on reasoning tasks. Although mathematical reasoning is a representative complex reasoning task, the generalizability of the method to other disciplines (such as physics or chemistry) or other types of reasoning tasks has not been verified.

**Questions:**

n/a

---

> ### Author Response · Authors · 2025-11-27
>
> We really thank and appreciate the Reviewer for the careful reading of our manuscript and the constructive comments. We are glad that you found the plug-and-play nature of GMTS within existing RLVR frameworks and affirm the contribution of our findings.
>
> > **Q1:** The idea of using gradients to identify important words is rather common and has been explored in various previous studies.
>
> **A1:**
> Thanks for your comments. We are pleased to clarify the core contributions of our work. Please refer to **Summary Part 1** for more details.
>
> > **Q2:** The evaluation is restricted to models under 8B parameters, making it unclear whether the proposed GMTS can generalize to larger and more capable LLMs.
>
> **A2:**
> Thanks for your comments. Due to computational resource limitations, we are currently unable to provide results on substantially larger models within the rebuttal period. However, our existing results already indicate a clear trend: as shown in **Figure 5 (Right)**, the performance gains of GMTS become more pronounced on stronger models. This trend is further supported by our newly added experiments on **commonsense_qa**, where GMTS achieves larger improvements over ETS when applied to stronger backbones, consistent with the behaviour observed in our math benchmarks. For a consolidated description of the newly added experiments and results, we kindly refer the reviewer to the **Summary Part 2, and 3** in our response.
>
> > **Q3:** The experimental gains reported in Tables 1 and 2 are modest, raising concerns about the practical impact of the proposed method.
>
> **A3:**
> Thank you for your comments. We have addressed this question in **Summary Part 4**; please kindly refer to that part for our detailed response.
>
> > **Q4:** The experiments primarily focus on reasoning tasks. Although mathematical reasoning is a representative complex reasoning task, the generalizability of the method to other disciplines (such as physics or chemistry) or other types of reasoning tasks has not been verified.
>
> **A4:**
> Thanks for your insightful suggestions. In response, we have added experiments on a different type of reasoning task: **commonsense_qa**, where we train **Qwen2.5-1.5B** and **Qwen2.5-7B** under all three methods. These results and discussion can be found in **Summary Part 2 and 3**. The new results show that GMTS still maintains a consistent advantage over the baselines on this benchmark. This suggests that the benefits of GMTS are not limited to mathematical reasoning but can also transfer to broader reasoning tasks. For physics and chemistry reasoning tasks, we were unable to identify suitable publicly available datasets for training within the scope of this work. We would greatly appreciate any pointers to such datasets.
>
> We sincerely thank the reviewer for the valuable comments. We hope that our responses and additional experiments help address these concerns.

---

### Author Response · Authors · 2025-11-27
**Summary of Rebuttal.**

We sincerely thank all reviewers for their thorough and thoughtful reviews. Each reviewer has provided valuable comments and suggestions that helped us improve the manuscript. We summarise and respond to the main points below.
### 1. Our main Contributions
(1) As illustrated in **Figure 2** in our manuscript, we observe an approximately linear correlation between $||\nabla\_\theta \log \pi\_\theta (o\_{i,t} |\boldsymbol{q},\boldsymbol{o}\_{i,<t})||$  and token entropy $E(o_{i,t})$, this relationship helps to justify **why entropy is important in RLVR**: for tokens in the same answer, high-entropy tokens tend to have larger gradient magnitude and therefore exert a stronger influence on the model updates.

(2) However, directly using gradient magnitude to measure token importance is hard because computing the token-level gradient is computationally prohibitive during training. To address this, we propose a novel metric, $\delta\_{i,t} = |E(o\_{i,t}) w\_{i,t} |$, which serves as a computationally efficient proxy for true token gradient magnitude. Both the entropy and coefficient $w\_{i,t}$ are already available as byproducts of standard RLVR training, making $\delta\_{i,t}$ efficient to compute and easy to integrate without significant overhead. Therefore, our approach offers a new perspective and a complementary technique on top of existing gradient-based methods, rather than one that conflicts with prior work.

We note that Reviewer X1D1, Reviewer P374, and Reviewer S8Hs expressed concerns about the performance of GMTS on tasks beyond math reasoning, and that Reviewer X1D1, Reviewer DEn5, Reviewer S8Hs, and Reviewer Q1A3 asked about its behaviour on larger or stronger models. We are very grateful for these constructive comments. We therefore conduct additional experiments, still using **avg@16** as the evaluation metric.

### 2. Other tasks
We conducted additional experiments as suggested by Reviewer P374. Specifically, on the **commonsense_qa** train dataset, we used **Qwen2.5-1.5B** and **Qwen2.5-7B** as base models for training. For evaluation, we evaluated these models on the **commonsense_qa (CS_QA)** [1] test dataset and the **commonsense_qa_2.0 (CS_QA2)** [2] dataset.

|**Qwen2.5-1.5B**|**CS_QA**|**CS_QA2**|**Average** |
|-|-|-|-|
|DAPO|77.22|52.81|65.01|
|+ETS Top (20%)|78.39|52.88|65.64|
|**+GMTS Top (20%)**|**79.35**|**53.00**|**66.12**|

| **Qwen2.5-7B**|**CS_QA**|**CS_QA2**|**Average** |
|-|-|-|-|
|DAPO|84.20|66.90|75.55|
| +ETS Top (20%)|83.10|66.20| 74.65|
| **+GMTS Top (20%)**|**85.65**|**67.24**|**76.45**|

### 3. Stronger base model.
Reviewer S8Hs suggested using more stronger model and training on more challenging tasks. In response, we added experiments with stronger 1.5B and 7B models: **DeepSeek-Qwen2.5-1.5b-Distill** and **DeepSeek-Qwen2.5-7b-Distill** trained on the Reasoning360 MATH dataset [3]. Due to limited computational resources and the time constraints of the rebuttal phase, we are unable to provide results on substantially larger models (e.g., 14B, 32B) within the rebuttal period. Our training is on 20% of the training data (11K examples) with the max response length of 5120 and evaluated on three difficult math tasks: **AIME2024, AIME2025 and AMC23**.

| **DeepSeek-Qwen2.5-1.5b-Distill** | **AIME24** | **AIME25** | **AMC23** | **Average** |
|-|-|-|-|-|
|DAPO (5K)|19.17|16.88|53.40|29.81|
|+ETS Top (20%)|18.13|18.33|55.20|30.55|
|**+GMTS Top (20%)**|**19.79**|**18.54**|**57.15**|**31.83**|

| **DeepSeek-Qwen2.5-7b-Distill** | **AIME24** | **AIME25** | **AMC23** | **Average** |
|-|-|-|-|-|
|DAPO (5K)|34.22|27.88|72.60|44.90|
|+ETS Top (20%)|35.60|29.13|74.15|46.29|
|**+GMTS Top (20%)**|**36.28**|**30.06**|**75.45**|**47.26**|

These additional results show that using a stronger math model on a more general dataset or switching to a non-math reasoning benchmark, GMTS still consistently outperforms ETS. For even larger models (**>8B**), we provide an anonymous **GitHub repository** with full training and evaluation code, linked on the first page of the revised manuscript.

### 4. Clarification on the Performance Improvements of GMTS.
We also note that Reviewer X1D1, Reviewer S8Hs, and Reviewer Q1A3 raised concerns regarding the performance improvements brought by GMTS.  On the challenging **AIME2024** and **AIME2025** benchmarks, GMTS shows clear gains over ETS: GMTS achieves **avg@16** performance of **39.79** and **30.00**, respectively, compared to 34.58 and 26.25 for ETS (i.e., **+5.21** and **+3.75** absolute points).  These consistent improvements on harder math problems with a stronger model suggest that GMTS has even greater potential when applied to larger and more capable LLMs. Also in **Figure 5 (Right)**, the performance gains of GMTS become more pronounced on stronger models.

[1] https://huggingface.co/datasets/tau/commonsense_qa

[2] https://huggingface.co/datasets/tasksource/commonsense_qa_2.0

[3] https://github.com/LLM360/Reasoning360

---

### Author Response · Authors · 2025-12-02
**Comments to AC, SAC,and PC**

Dear ICLR AC, SAC, and PC members:

Following the recent incident in the ICLR review process, we recognize that the additional workload has been created for you and are grateful for your efforts to ensure a fair and rigorous evaluation of all submissions. To help reduce this burden, we provide a three-part overview of Reviewers' concerns and how we address them.

### (1) Main questions and our responses

>  **Reviewer X1D1 and Reviewer S8Hs** raised questions about our contributions and asked us to clarify in more detail how GMTS differs from ETS, beyond simply introducing an additional weighting term.

We restate our main contributions and clarify the distinctions between GMTS and ETS in **Part 1** of the **Summary of Rebuttal**.

>  **Reviewer X1D1, Reviewer P374, and Reviewer S8Hs** would like to see how GMTS performs on reasoning tasks beyond the MATH-style datasets.

We conducted additional experiments. The detailed results and settings for these experiments are provided in **Part 2** of the **Summary of Rebuttal**.

>   **Reviewer X1D1, Reviewer DEn5, Reviewer S8Hs, and Reviewer Q1A3** asked about the behaviour of GMTS on larger (e.g., 14B, 32B) or stronger models.

Due to resource limitations, we are unable to run experiments on larger models (e.g., 14B or 32B) during the rebuttal period. However, following the suggestion of **Reviewer S8Hs**, we conducted additional experiments on stronger models, **DeepSeek-Qwen2.5-1.5b-Distill** and **DeepSeek-Qwen2.5-7b-Distill**. The detailed results are reported in **Part 3** of the **Summary of Rebuttal**.

>  **Reviewer X1D1, Reviewer S8Hs, and Reviewer Q1A3** raised questions regarding the performance improvements of GMTS.

The detailed responses to this question are reported in **Part 4** of the **Summary of Rebuttal**.

### (2) Clarifications in the revised manuscript

> **Reviewer DEn5**  asked for the code.

We have provided our code in the revised version.

> **Reviewer P374** and **Reviewer S8Hs** asked for the examples,  **Reviewer Q1A3** asked for the correlation plots between $E(o_{i,t}) $ and $E(o_{i,t}) * \omega_{i,t}$

We provide qualitative examples illustrating the differences between ETS and GMTS in terms of the tokens selected under **low-advantage** and **high-advantage** settings. We have added the correlation plots between  $E(o_{i,t}) $ and $E(o_{i,t}) * \omega_{i,t}$ to further support our analysis.

### (3) Other comments from Reviewers

Apart from the points discussed above, we have also carefully addressed the other types of comments raised by the reviewers. We clarified the evaluation settings and hyperparameters (**Reviewer DEn5**), provided a more detailed explanation of the formulas (**Reviewer P374**), and explained convergence and additional advantages of GMTS (**Reviewer Q1A3**). These issues have all been addressed in the discussions.



We are especially grateful to the AC for the additional time and care devoted to our submission.  We hope these clarifications and results highlight the substantive improvements of GMTS over ETS and its potential for broader tasks and stronger models.

-- **All authors of manuscript 25206**

---

### Meta-Review · Area_Chair_Nhf9 · 2025-12-23

**Summary:**

This paper proposes a token selection method named GMTS to enhance RLVR's performance on reasoning tasks. Five reviewers submitted comments, to which the authors provided rebuttals; most reviewers did not engage in the discussion. The final score approached the borderline, with four reviewers leaning toward rejection. After carefully reviewing the paper and author responses, the AC noted that some reviewer concerns remained unaddressed, such as the method's limited novelty and modest performance gains. The AC recommended that the authors revise and submit to a subsequent conference.

**Reviewer Concerns:**

I believe that some issues concerning methodological innovation remain unresolved.

**Reviewer Scores:**

I believe the reviewers will maintain the original score.

---

### Decision · Program_Chairs · 2026-01-26

Reject